# Identification of the periplasmic DNA receptor for natural transformation of *Helicobacter pylori*

Prashant P. Damke[1,5], Anne Marie Di Guilmi[1], Paloma Fernández Varela [2], Christophe Velours[2], Stéphanie Marsin[2], Xavier Veaute [3], Mérick Machouri[1], Gaurav V. Gunjal[4], Desirazu N. Rao[4], Jean-Baptiste Charbonnier[2] & J. Pablo Radicella [1]*

Horizontal gene transfer through natural transformation is a major driver of antibiotic resistance spreading in many pathogenic bacterial species. In the case of Gram-negative bacteria, and in particular of *Helicobacter pylori*, the mechanisms underlying the handling of the incoming DNA within the periplasm are poorly understood. Here we identify the protein ComH as the periplasmic receptor for the transforming DNA during natural transformation in *H. pylori*. ComH is a DNA-binding protein required for the import of DNA into the periplasm. Its C-terminal domain displays strong affinity for double-stranded DNA and is sufficient for the accumulation of DNA in the periplasm, but not for DNA internalisation into the cytoplasm. The N-terminal region of the protein allows the interaction of ComH with a periplasmic domain of the inner-membrane channel ComEC, which is known to mediate the translocation of DNA into the cytoplasm. Our results indicate that ComH is involved in the import of DNA into the periplasm and its delivery to the inner membrane translocator ComEC.

[1] Institute of Cellular and Molecular Radiobiology, Institut de Biologie François Jacob, CEA, Université de Paris and Université Paris Sud, F-92265 Fontenay aux Roses, France. [2] Institute for Integrative Biology of the Cell (I2BC), CEA, CNRS, Univ. Paris-Sud, Université Paris-Saclay, F-91198 Gif-sur-Yvette, France. [3] Institute of Cellular and Molecular Radiobiology, Institut de Biologie François Jacob, CEA, INSERM, Universités Paris Diderot and Paris Sud, F-92265 Fontenay aux Roses, France. [4] Department of Biochemistry, Indian Institute of Science, Bangalore 560012, India. [5] Present address: Gluck Equine Research Center, University of Kentucky, Lexington, KY 40546, USA. *email: pablo.radicella@cea.fr

Horizontal gene transfer (HGT) is a key driver of bacterial evolution. In pathogenic bacteria HGT favours the adaptation to new hosts and the spread of virulence factors and antibiotic resistance traits. One of the major modes of HGT in many bacterial species is natural transformation (NT), a pathway by which bacteria import DNA from the environment and internalise it into the cytoplasm where it can integrate into the bacterial genome. DNA uptake and integration by naturally transformable bacterial species is a complex process involving a dedicated set of proteins[1,2]. Unlike other HGT mechanisms such as transduction or conjugation, NT relies exclusively on proteins coded by the recipient cell.

In Gram-negative bacteria, where the incoming DNA has to pass through two membranes and the periplasmic space in between, work on *Helicobacter pylori*[3,4], *Neisseria gonorrhoeae*[5] and *Vibrio cholerae*[6,7] showed that DNA transport through the bacterial envelope is a multi-step process. After binding to the cell surface, double-stranded DNA (dsDNA) is internalised into the periplasm. With the exception of *H. pylori*, both Gram-positive and Gram-negative transformable bacteria use a (pseudo-) pilus similar to Type IV pili as DNA import machinery[1,8,9]. However, these (pseudo-) pili are not sufficient for the transport of DNA through the outer membrane. Indeed, the protein ComE(A), also conserved amongst most naturally transformable Gram-positive and Gram-negative bacteria, is required for DNA uptake[1,4,5,10–12]. In *V. cholerae* and *N. gonorrhoeae*, ComE(A) is the periplasmic DNA receptor[6,10]. Once in the periplasm, the transforming DNA (tDNA) is delivered to the inner membrane transporter ComEC[13]. During this step, the dsDNA is processed to yield a single-stranded DNA (ssDNA) suitable for its ComEC-mediated translocation into the cytoplasm. The periplasmic proteins responsible for processing the tDNA and delivering it to ComEC are yet to be defined in any Gram-negative transformable species. Finally, through a still unknown mechanism, ComEC allows the ssDNA to cross the inner membrane and reach the cytoplasm.

*H. pylori* infection of the human stomach triggers a chronic gastritis that can evolve into a series of severe pathologies such as gastroduodenal ulcers and cancer[14,15]. Infecting about half of the world population, *H. pylori* is one of the most successful bacterial pathogens. Its amazing genetic diversity and variability are certainly major contributors to this success by allowing the emergence of new alleles. Furthermore, the new alleles, as well as antibiotic resistance genes, can rapidly propagate through natural transformation, a very efficient mechanism in *H. pylori*[16–21]. Amongst naturally transformable bacteria, *H. pylori* is unique and divergent in terms of the composition of its competence machinery. Rather than a canonical Type IV (pseudo-) pilus, *H. pylori* employs a type-IV secretion system (T4SS), ComB, for initial DNA uptake during transformation[22,23]. More surprisingly, no orthologue of the conserved ComE(A) DNA receptor has been identified[24], raising the question of how tDNA is imported into the periplasm. A recent study showed that *H. pylori* import large amounts of tDNA into its periplasm[25], suggesting the presence of an efficient DNA receptor protein in that bacterial compartment.

In this study, we identify ComH, a previously uncharacterised *H. pylori* protein, as the periplasmic DNA receptor essential for NT. We show that ComH is required for the transfer of external DNA to the periplasm. ComH interacts with DNA through its C-terminal domain of unknown fold and with ComEC putative oligonucleotide binding (OB) fold through its N-terminal domain, providing a carrier for the transforming DNA between the outer and inner membranes.

## Results

**ComH is essential for the import of transforming DNA.** *comH* was originally identified as a gene required for NT by screening a mutant library of *H. pylori* for non-transformable strains[26]. We confirmed the effect of *comH* inactivation on NT by determining the frequencies of integration of a streptomycin resistant (StrepR) marker using as donor total genomic DNA from a StrepR isogenic strain. When compared to the *wild-type* strain, the Δ*comH* mutant displayed >10,000-fold reduction in the yield of recombinant clones, an effect similar to that resulting from inactivation of *comB2* or *comEC* (Table 1), both essential for natural competence[22,23,27]. The values obtained are however slightly higher than the spontaneous mutation frequencies, $6.3 \times 10^{-9}$ and $1.09 \times 10^{-9}$ for the *wild type* and *comH* strains, respectively, suggesting that some transformation is still taking place. To complement the mutant strain we inserted in *rdxA*, a non-essential locus, the ComH-FLAG coding sequences fused to the *comH* upstream sequences and the N-terminal signal peptide from ComH. Complementation by the ectopically expressed ComH protein restored the yield of recombinants to levels comparable to those of the *wild-type* strain (Table 1), ruling out the possibility of a polar effect in the Δ*comH* strain.

The transformasome of *H. pylori* is composed of proteins involved in either tDNA uptake and transport across the bacterial envelope (*i.e.* components of T4SS and ComEC) or in its handling within the cytoplasm leading to chromosomal integration (*i.e.* DprA, RecA)[28,29]. To define in which of those two steps ComH is required, electroporation of different *H. pylori* mutant strains with a 139-mer chemically synthesised ssDNA coding for streptomycin resistance was performed and the recombinant frequencies were determined. ssDNA is a very poor substrate for natural transformation[30]. We did not observe significant number of StrepR colonies using the standard transformation protocol with the ssDNA but electroporation with the same substrate yielded a recombination frequency of $5.28 \times 10^{-7}$ (Table 2). Delivery of the ssDNA into the cytoplasm by electroporation should allow circumventing the initial binding and transport processes. Indeed, recombination frequencies obtained by electroporation with ssDNA of the Δ*comB2* and Δ*comEC* strains

**Table 1 Natural transformation frequencies for *H. pylori* strains.**

| Genotype | Mean transformation frequencies | n | Relative value | P-value (MWU) |
|---|---|---|---|---|
| *wild-type* | $1.60 \times 10^{-3}$ ($1.87 \times 10^{-3}$) | 27 | 1.00 | |
| Δ*comH* | $7.07 \times 10^{-8}$ ($1.26 \times 10^{-7}$) | 11 | $4.42 \times 10^{-5}$ | <.0001 |
| Δ*comH rdxA::comH-FLAG* | $5.54 \times 10^{-4}$ ($4.93 \times 10^{-4}$) | 9 | 0.35 | 0.0356 |
| Δ*comB2* | $6.68 \times 10^{-9}$ ($4.34 \times 10^{-9}$) | 4 | $4.17 \times 10^{-6}$ | <.0001 |
| Δ*comEC* | $2.45 \times 10^{-9}$ ($4.90 \times 10^{-9}$) | 4 | $1.53 \times 10^{-6}$ | <.0001 |
| Δ*dprA* | $3.97 \times 10^{-6}$ ($1.69 \times 10^{-6}$) | 4 | $2.48 \times 10^{-3}$ | <.0001 |
| Δ*recA* | $4.05 \times 10^{-9}$ ($2.32 \times 10^{-9}$) | 3 | $2.53 \times 10^{-6}$ | <.0005 |

The recombination frequencies of the isogenic streptomycin resistant (StrepR) total genomic DNA were calculated as the number of streptomycin resistance colonies per recipient colony-forming unit. Values correspond to the mean and standard deviation. *n* No. of independent determinants. *MWU* Mann–Whitney U test.

**Table 2 Transformation frequencies determined for electroporated *H. pylori* strains.**

| Genotype | Mean recombination frequencies | *n* | Relative value | *P*-value (MWU) |
|---|---|---|---|---|
| *wild-type* | $5.28 \times 10^{-7}$ ($5.86 \times 10^{-7}$) | 16 | 1.00 | |
| $\Delta comH$ | $3.25 \times 10^{-7}$ ($1.83 \times 10^{-7}$) | 5 | 0.61 | 0.9555 |
| $\Delta comB2$ | $1.90 \times 10^{-6}$ ($1.48 \times 10^{-6}$) | 4 | 3.59 | 0.1482 |
| $\Delta comEC$ | $4.36 \times 10^{-7}$ ($5.95 \times 10^{-7}$) | 9 | 0.82 | 0.4605 |
| $\Delta dprA$ | $1.54 \times 10^{-8}$ ($2.14 \times 10^{-8}$) | 5 | 0.029 | <.0001 |
| $\Delta recA$ | 0 | 4 | 0 | 0.0004 |

Exponentially growing *H. pylori* cells were electroporated with 139-mer single-stranded DNA coding for streptomycin resistance and the recombination frequencies were calculated as the number of streptomycin resistance colonies per recipient colony-forming units. Values correspond to the mean and standard deviation. *n* No. of independent determinants. *MWU* Mann–Whitney U test.

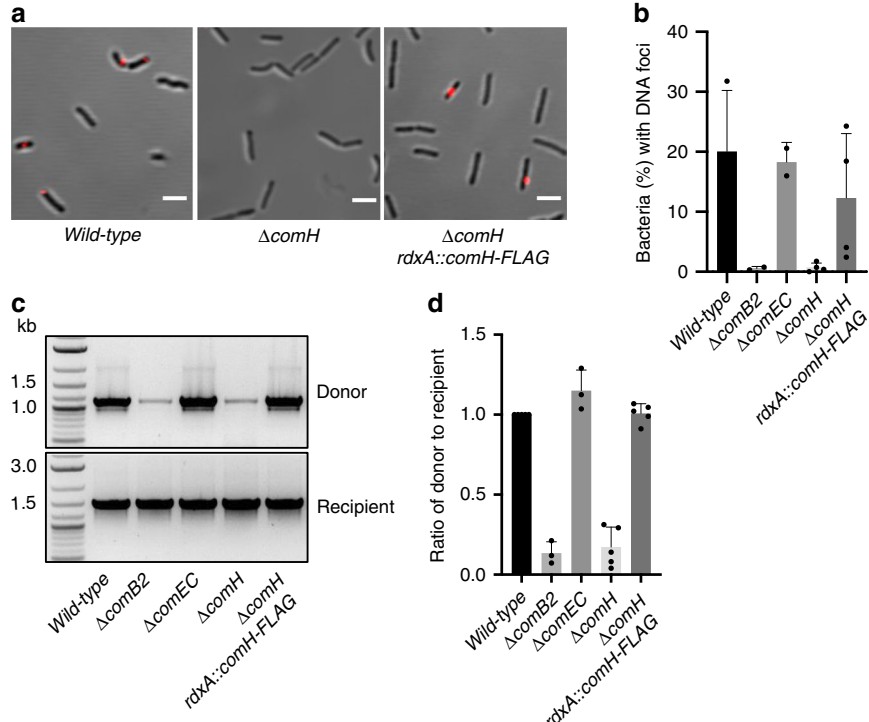

**Fig. 1** ComH is essential for tDNA import into the periplasm. **a** Fluorescent DNA foci formation in *H. pylori* wild-type, Δ*comH* and Δ*comH redxA::comH-FLAG* strains. Z maximum projections of merged images of ATTO-550 (red channel) and differential interference contrast (DIC) are presented. Scale bar = 2.5 μm. **b** Percentage of bacteria with fluorescent DNA foci from the experiments described in **a**. The mean and standard deviation values calculated from at least two independent experiments are shown for wild-type (*n* = 661), Δ*comB2* (*n* = 633), Δ*comEC* (*n* = 1068), Δ*comH* (*n* = 989) and Δ*comH rdxA:: comH-FLAG* (*n* = 5245) cells. Bars correspond to the mean+/- SD. **c** The presence of donor DNA in the periplasm was monitored by PCR. The products of amplification were visualised by agarose gel electrophoresis. **d** The ratios of donor versus recipient amplification were obtained by scanning of the agarose gels using ImageJ and the results were normalised with respect to the *wild-type* strain. Bars correspond to the mean of at least 3 independent experiments+/- SEM.

were similar to that of the wild-type strain, while almost undetectable for the Δ*dprA* and Δ*recA* strains (Table 2). Electroporation of the Δ*comH* strain with ssDNA allowed recombination frequencies close to that of the wild-type strain (Table 2). Taken together these results show that ComH, like the ComB complex and ComEC, is involved in the delivery of the tDNA into the cytoplasm but dispensable for its integration into the chromosome.

**ComH is required for the import of tDNA into the periplasm.** During NT in *H. pylori*, fluorescently labelled tDNA can be transiently detected in the periplasm as discreet foci in *wild-type* strains but not in those mutated in *comB*[4,31]. To better define the role of ComH, we determined the efficiency of tDNA foci formation (Fig. 1a). While ~20% of the *wild-type* bacteria displayed

fluorescent tDNA foci, <1% of the Δ*comB2* cells had detectable foci (Fig. 1b). In a Δ*comEC* strain, the proportion of bacteria with tDNA foci was similar to that of the *wild-type* (Fig. 1b), consistently with the role of ComEC in the DNA transport through the inner membrane[4]. When ComH was disabled, only 1% of the cells presented tDNA foci. Wild-type levels were recovered in the Δ*comH* strain by ectopic expression of ComH-FLAG (Fig. 1a, b).

We also monitored by PCR the presence of internalised DNA in the different mutant strains (Fig. 1c, d). After incubation of the cells in the presence of a 6.3 kb linear dsDNA with no homology to the *H. pylori* genome, the extracellular and unprotected DNA was removed by extensive washing and nuclease treatment. After isolating total DNA, a 1.1 kb region of the internalised donor dsDNA was amplified by PCR. In agreement with the formation of tDNA foci, amplification of internalised tDNA was observed in the *wild-type* while only residual amplification was detected in a

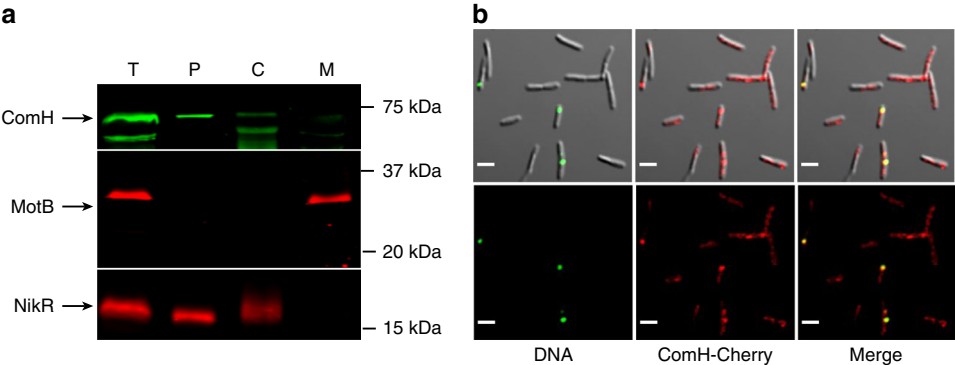

**Fig. 2** ComH is present in the periplasm where it co-localises with tDNA foci. **a** ComH-Flag was ectopically expressed in a Δ*comH* strain and the presence of the protein in the different subcellular compartments was detected by immunoblotting. T: Total extract, P: periplasm, C: cytoplasm, M: membrane. MotB and NikR were used as controls for the fractionation experiments. **b** Co-localisation of ComH-mCherry expressed in a Δ*comEC* strain from its own locus with ATTO-488-dUTP labelled transforming DNA. Z maximum projections of merged and separate images of ATTO-488 (green channel), ComH-mCherry (red channel), and differential interference contrast are presented. Scale bar = 2.5 μm.

Δ*comB2* strain (Fig. 1c, d). Because the level of amplified tDNA in a Δ*comEC* strain was similar to that of the *wild-type*, it can be assumed that it essentially reflects the DNA present in the periplasm (Fig. 1c, d). Disruption of *comH* led to a dramatic reduction in the levels of internalised DNA, comparable to those of the Δ*comB2* strain, that could be reverted by the ectopic expression of the gene in the Δ*comH rdxA::comH-FLAG* strain (Fig. 1c, d). The residual amplification observed in the competence mutant strains is likely due to non-specific DNA binding to the cell surface. Collectively, the foci analysis and the PCR amplification experiments showed that ComH is essential for the presence of tDNA in the periplasm during natural transformation.

**ComH is in the periplasm where it co-localises with tDNA foci**. We then proceeded to determine the subcellular localisation of ComH by fractionation of exponentially growing *H. pylori* Δ*comH* cells expressing a FLAG-tagged version of the protein. Western blot analysis showed that the majority of ComH-FLAG was present in the periplasm (Fig. 2a) consistently with the presence of a signal peptide (amino acids 1–19) (http://www.cbs.dtu.dk/services/SignalP/)[32] potentially targeting the protein to this compartment. With the fractionation conditions used in this experiment no association of ComH with the membrane fraction was detected. The presence of ComH in the periplasm was confirmed using a strain expressing a HA-tagged version of the protein from the *comH* locus (Supplementary Fig. 1).

In view of its localisation and its role in the import of tDNA into the periplasm, we asked whether ComH co-localised with the tDNA during natural transformation. To address this question, we generated a *H. pylori* strain expressing a ComH-mCherry fusion protein yielding transformation frequencies comparable to those of the *wild-type* strain (Supplementary Table 1). The expression and stability of ComH-mCherry in *H. pylori* was verified by Western blot (Supplementary Fig. 2). Confocal microscopy analysis revealed that the majority of the cells showed a rather homogenous and weak fluorescent signal on which interspaced puncta or clusters could be observed (Fig. 2b). This pattern was observed independently of the genetic background. The number of clusters increased with time, reaching a maximum 90 min after putting the cells in contact with DNA. When a *comEC* disrupted strain (*comH-mCherry* Δ*comEC*) was used for the co-localisation experiment in order to stabilise the tDNA foci, we found that 423 out of the 553 tDNA foci analysed (76%) co-localised with ComH-mCherry clusters (Fig. 2b). This was confirmed by a high Manders' coefficient (0.88 ± 0.02)

reflecting the fraction of green signal (DNA foci) with the clusters of red signal (ComH-Cherry) determined using a threshold defined by the JACOP plugin from ImageJ[33]. The detection of ComH clusters without DNA foci could reflect the presence of non-labelled DNA arising from bacterial lysis. This is consistent with the fact that genetic markers can be transferred from one strain to another by simply co-culturing them.

**ComH binds double-stranded DNA**. Despite the fact that no obvious DNA binding motif could be identified by sequence analysis of the protein, the co-localisation of ComH with tDNA and the inability of Δ*comH* strain to promote tDNA entry led us to hypothesise that ComH could be the periplasmic receptor for the incoming DNA. While in other naturally transformable Gram-negative bacteria this is the role of ComE(A)[5,6], no orthologue of such proteins has been identified in *H. pylori*. We therefore tested the *in vitro* DNA binding ability of purified ComH. We first assessed the DNA binding capacity of the purified C-terminally His-tagged fusion protein (ComH-His$_6$) (Supplementary Fig. 3a) by electrophoretic mobility shift assays (EMSA). ComH-His$_6$ formed discrete nucleoprotein complexes with an 18 bp dsDNA in a concentration dependent manner while no binding to ssDNA (18-mer) was detectable (Fig. 3a). The same preference for dsDNA over ssDNA was observed when Surface Plasmon Resonance (SPR) was used to characterise the DNA interaction properties of the ComH-His$_6$. The binding sensorgrams show that ComH-His$_6$ protein bound dsDNA with high affinity while only residual binding was detected when ssDNA was used as substrate (Fig. 3b). Neither the Langmuir isotherm nor the equilibrium approach using the Proteon Manager software gave satisfactory fittings for the calculation of affinity constants. In order to obtain quantitative data on the binding affinity of ComH for dsDNA, we used the microscale thermophoresis (MST) using a 18 bp dsDNA labelled with a 5'FAM[34]. The observed $Kd$ was 0.05 ± 0.03 μM (Fig. 3c). We also analysed the interaction between ComH and 18 bp dsDNA or 18nt ssDNA using sedimentation velocity analytical ultracentrifugation (SV-AUC). With the dsDNA we observed the formation of a major complex with a sedimentation coefficient (6.5S). and with a Kd of 0.1 ± 0.2 μM (Fig. 3 d and Supplementary Fig. 4a). No complex was observed in the same conditions with the ssDNA (Supplementary Fig. 4b).

To determine the DNA binding characteristics of ComH we used the purified MBP-ComH protein (Supplementary Fig. 3b) in EMSA. MBP-ComH showed affinity for a variety of DNA substrates, as long as they harboured a double-stranded region,

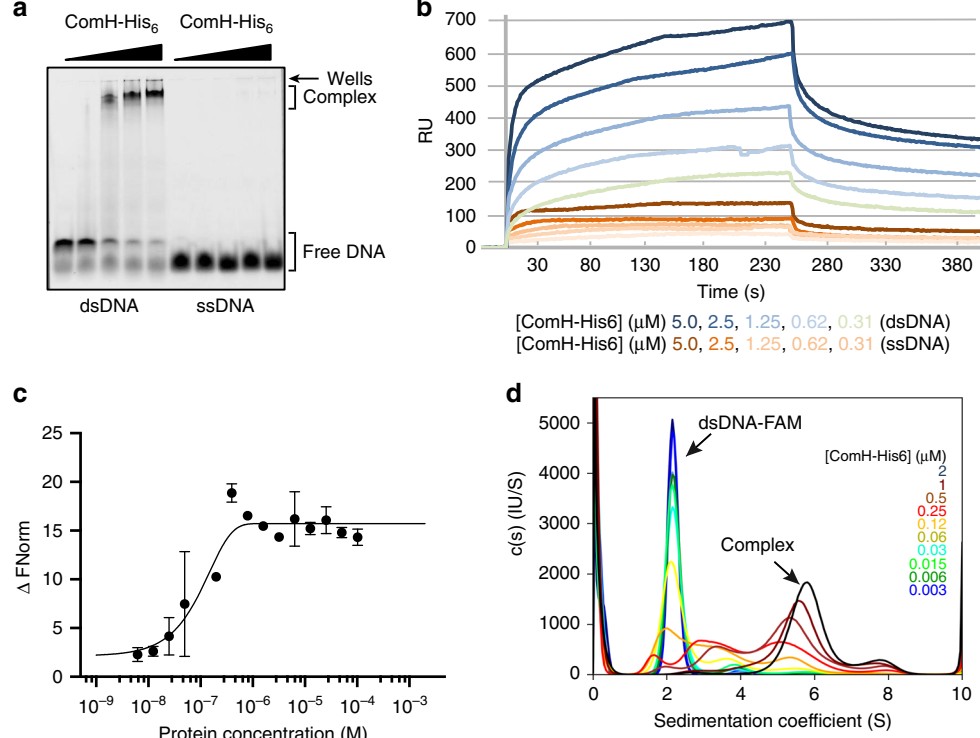

**Fig. 3** ComH is a double-stranded DNA- binding protein. **a** Visualisation by electrophoretic mobility shift assays of nucleoprotein complexes formed by ComH-His$_6$. Increasing concentrations of purified ComH-His$_6$ (0, 0.5, 1, 1.5, and 2 μM) were incubated with Cy5-labelled dsDNA (18 bp) and ssDNA (18-mer) substrates (30 nM). The free DNA substrates and nucleoprotein complexes were resolved by native PAGE (6%). **b** Comparison of ComH-His$_6$ binding to dsDNA (50 bp) and ssDNA (50-mer) by Surface Plasmon Resonance. Increasing concentrations of ComH-His$_6$ (0, 0.31, 0.62, 1.25, 2.5, and 5 μM) in binding buffer (PBS + 0.005% Tween20) were injected onto the streptavidin chip containing immobilised DNA substrates. The SPR derived sensograms for dsDNA and ssDNA are shown. **c** Interaction measurements by microscale thermophoresis of the affinity between ComH-His$_6$ and a 18 bp DNA labelled with a FAM in 5'. All experiments were performed in duplicate and error bars = SD. **d** Analyses by sedimentation velocity analytical ultracentrifugation (SV-AUC) of the interaction between ComH and 18 bp dsDNA-FAM. The free dsDNA-FAM has a sedimentation coefficient of 2.1S. ComH forms a main complex with a sedimentation coefficient of 6.5S.

but not for ssDNA (Supplementary Fig. 5a). The nucleoprotein complexes formed by MBP-ComH with a 39 bp dsDNA were stable up to 600 mM of either NaCl or KCl (Supplementary Fig. 5b). Purified MBP alone did not display binding to tested dsDNA using this assay (Supplementary Fig. 5c).

Taken together, the results obtained using two different tagged constructs of ComH protein and four different experimental approaches, show that ComH is a novel DNA binding protein, with strong affinity for duplex DNA compared to ssDNA.

**ComH binds to dsDNA through its C-terminal domain.** The full length ComH protein is composed of 479 amino acids (54.78 kDa). Preliminary bioinformatics analysis of ComH predicted the presence of two structured regions at the N- and C-termini of the protein connected by a linker sequence. Neither region showed potential structural homology to other proteins and therefore they represent novel and uncharacterised domains of unknown function. To search for the ComH DNA-binding domain, we separately purified the N-terminal (ComH-NTD) (amino acids 20–169) and C-terminal domains (ComH-CTD) (amino acids 170–479) of the protein fused to either His$_6$ or MBP tags (Supplementary Fig. 3a, b).

We first tested the DNA binding characteristics of ComH-CTD-His$_6$ and ComH-NTD-His$_6$ using EMSA. As the full-length protein, ComH-CTD-His$_6$ bound dsDNA while it had essentially no-affinity for ssDNA (Fig. 4a, b). On the other hand, ComH-NTD-His$_6$ had basically no DNA binding capacity to either dsDNA or ssDNA in the same protein concentration range

(Fig. 4b, c). Similar results were obtained when we tested the DNA binding capacities of the MBP fused domains. Both, full-length MBP-ComH and MBP-ComH-CTD formed discrete nucleo-protein complexes when incubated in the presence of dsDNA while the MBP-ComH-NTD displayed negligible DNA binding affinity (Supplementary Fig. 6a, b). In order to obtain quantitative data on the relative binding affinity of full-length ComH versus the N- and C-terminal domains for dsDNA, the interactions of the His$_6$-tagged domains were analysed by MST. The $Kd$ obtained for ComH-CTD was $3.4 \pm 1.7$ μM (Fig. 4d). While this value reflects a lower affinity than the full-length protein, the ComH-NTD had a much weaker affinity ($Kd > 40$ μM) (Supplementary Fig. 6c) in agreement with the EMSA analyses. Finally, we addressed the question of whether ComH needed free DNA ends for loading using His$_6$ tagged constructs. Both, ComH-His$_6$ and ComH-CTD-His$_6$ displayed concentration dependent binding to linear and covalently closed supercoiled plasmid DNA irrespective of their topology (Supplementary Fig. 7a, b).

In conclusion, the DNA binding properties of the truncated version of ComH suggest that the CTD harbours a novel DNA binding motif responsible for the affinity of ComH for dsDNA.

**ComH-CTD is sufficient to allow tDNA entry into the periplasm.** Since the CTD of ComH is sufficient for DNA binding, we tested whether expression of this domain in a Δ*comH* strain was sufficient to revert the mutant phenotype. For this purpose, as we did for the full length protein, we inserted in *rdxA* the

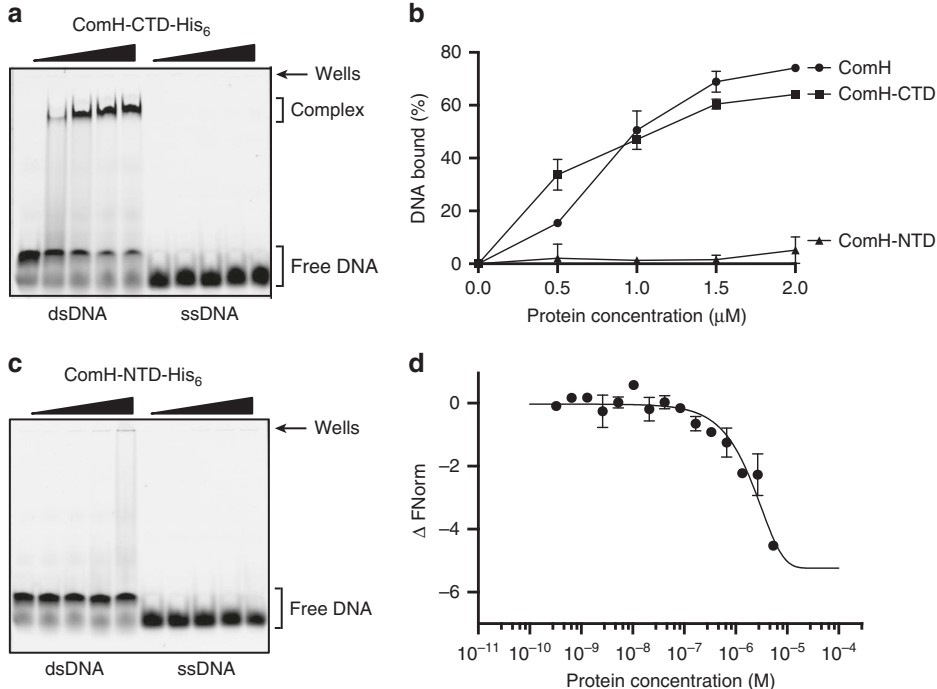

**Fig. 4** ComH binds to dsDNA through its C-terminus. EMSA analysis of DNA interaction by purified **a** ComH-CTD-His$_6$, and **c** ComH-NTD-His$_6$ with dsDNA (18 bp) and ssDNA (18-mer). Increasing concentrations of purified His$_6$ fused ComH domains (0, 0.5, 1, 1.5, and 2 μM) were tested for binding with indicated DNA substrates (30 nM). Representative images of resolved gels are shown. Quantification of the bands from Figs. 4a and 4c was performed using ImageJ and the percentage binding is displayed in **b**. Error bars represent standard deviation from at least two independent experiments. **d** Interaction measurements by microscale thermophoresis of the affinity between ComH-CTD-His$_6$ and a 18 bp DNA labelled with a FAM in 5′. All experiments were performed in duplicate. Error bars represent standard deviation from at least two independent experiments.

ComH-CTD-FLAG coding sequences fused to the *comH* upstream sequences and the N-terminal signal peptide from ComH (Fig. 5a). We then confirmed that ComH-CTD-FLAG was expressed and exported to the bacterial periplasm like the full-length form (Fig. 5b). The ectopic expression of ComH-CTD-FLAG resulted in a partial restoration of tDNA foci formation (Fig. 5c). The capacity of the CTD to allow the import of the tDNA to the periplasm was confirmed by the duplex PCR assay. Indeed, ectopic expression of ComH-CTD in the *ΔcomH* strain allowed significant amplification of the tDNA retained by the bacteria after nuclease treatment, albeit to lower levels than those observed for the mutant strain complemented with the full-length ComH (Fig. 5d).

We then assessed the capacity of the truncated form of ComH to allow the whole natural transformation process up to the chromosomal integration of the tDNA. Unlike that of the full-length ComH-FLAG, the ectopic expression of ComH-CTD-FLAG in the *ΔcomH* strain did not restore the capacity to obtain recombinants (Fig. 5e).

Taken together, these results suggest that while the CTD of ComH is sufficient for the normal detection of the tDNA in the periplasm of competent cells during NT, it does not support its translocation into the cytoplasm. These results were further confirmed following the internalisation of fluorescently labelled DNA into the bacterial cytoplasm, as previously described[31] (Supplementary Movies 1–3). While in a *wild-type* strain the volume of internalised DNA increased over time, in a *ΔcomEC* mutant the tDNA remained blocked in the periplasm, reflecting the impairment in this mutant of the transport through the inner membrane. When the *ΔcomH rdxA::comH-CTD-FLAG* strain was tested in this assay, it displayed a phenotype very similar to that of the *ΔcomEC* mutant (Supplementary Fig. 8).

**ComH NTD interacts with the periplasmic OB domain of ComEC.** The similitude of phenotypes between the *ΔcomEC* and *ΔcomH rdxA::comH-CTD-FLAG* strains (periplasmic foci formation but no translocation of the tDNA into the cytoplasm) raised the possibility that ComH might act in close association with ComEC to allow the transport of tDNA through the inner membrane. Topological prediction of *H. pylori* ComEC structure suggests that this inner membrane protein has 9 transmembrane segments and an oligonucleotide binding (OB) domain[35] present on the periplasmic side of the membrane (Fig. 6a). We therefore tested a potential interaction between ComH and the ComEC-OB domain. Extracts from *E. coli* bacteria expressing His$_6$-tagged recombinant fusions of the ComEC-OB domain were incubated with *E. coli* extracts containing either MBP or MBP-ComH. The products of amylose affinity pull-down from the mixtures were then eluted from the resin by competition with maltose and probed by immunoblot for the presence of the ComEC-OB domain using anti-His tag antibodies. As shown in Fig. 6b, ComEC-OB-His$_6$ was recovered from the pull-down when co-incubated with full-length MBP-ComH, indicating a direct interaction between ComH and the OB domain from ComEC. To rule out the possibility that the interaction was mediated by DNA present in the extracts, the pull-down experiments were repeated with the addition of benzonase. The interaction between ComH and the ComEC OB domain persisted even after degradation of the DNA (Supplementary Fig. 9a, b). Interestingly, the His$_6$-tagged OB domain was also recovered when MBP-ComH-NTD was used as bait but not in the case of MBP-ComH-CTD or MBP alone. These results showed that through its NTD, ComH is able to interact with ComEC, providing a link between the handling of tDNA in the periplasm and its passage through the inner membrane.

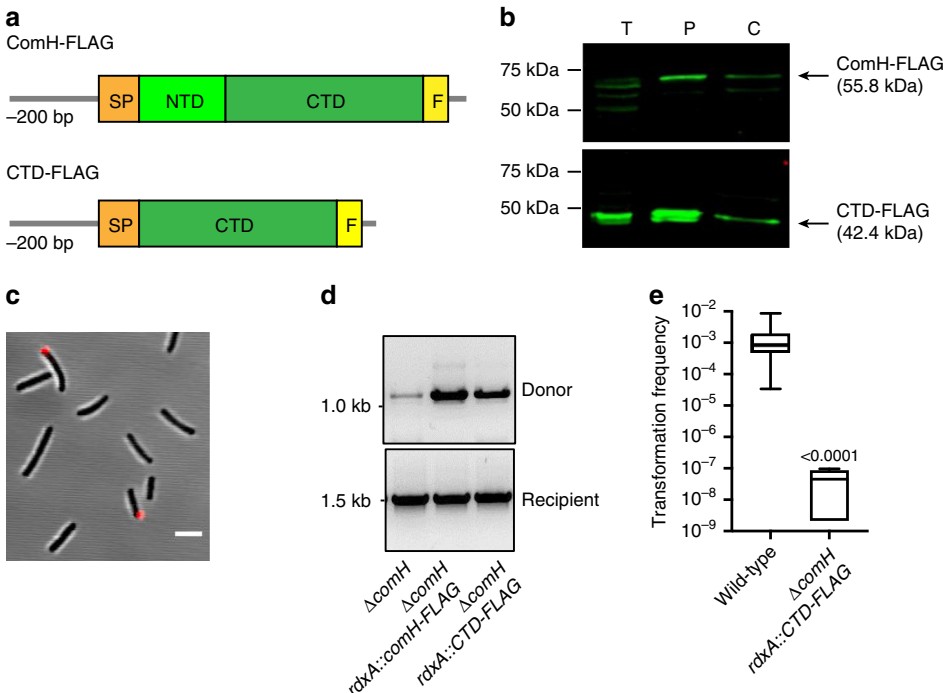

**Fig. 5** Ectopic expression of ComH DNA binding domain (ComH-CTD) restores DNA uptake but not transformation. **a** Schematic representation of ectopically expressed FLAG-tagged ComH constructs in a *ΔcomH* genetic background. The upstream sequence (-200 bp) and the coding sequence for the signal peptide of *comH* (SP) were added to maintain the expression levels of the gene close to that of the *wild-type* strain. **b** Subcellular localisation of ComH-FLAG and ComH-CTD-FLAG was monitored by immunoblotting using anti-FLAG antibodies. T: Total extract, P: periplasm, C: cytoplasm. **c** Fluorescent DNA (ATTO-550-dUTP labelled 408 bp dsDNA) foci formation in the *ΔcomH rdxA::comH-CTD-FLAG* strain. 3394 cells were counted from at least two independent experiments. 256 (7.5%) bacteria showed presence of detectable DNA foci. Scale bar = 2.5 μm. **d** Detection of internalised donor DNA in the *ΔcomH rdxA::comH-CTD-FLAG* strain by the PCR assay. **e** Comparison of recombination frequencies after natural transformation with genomic DNA of *wild-type* and *ΔcomH rdxA::comH-CTD-FLAG* strains. The central line represents the median, the bounds of the boxes show the inner quartile range and the whiskers represent the minimum and maximum values of the data set. The numbers above the box refer to P-values calculated using nonparametric MWU T tests.

## Discussion

During natural transformation *H. pylori* displays a remarkable efficiency to import tDNA into the periplasm of the bacteria and its subsequent recombination in the genome[25,36]. Pioneering work in *H. pylori* showed that this pathogen uses a two-step DNA uptake mechanism in which first the dsDNA is transported from the surface of the cell to the periplasm and subsequently it is transferred to the cytoplasm through the inner membrane[4]. Similar two-step DNA uptake mechanisms were described for the Gram-negative bacteria *N. gonorrhoeae* and *V. cholerae*[4,5,10]. While the first step is mediated by species-specific DNA uptake machineries, the ComB type IV secretion system in the case of *H. pylori* or type IV (pseudo-) pili for the other naturally transformable bacteria, the second step of DNA translocation across the cytoplasmic membrane is mediated by the conserved ComEC (ComA in *Neisseria*) channel[3]. In *N. gonorrhoeae* and *V. cholerae* it is proposed that the periplasmic DNA binding protein ComEA (ComE in *Neisseria*) is required for DNA traversal of the outer membrane by retaining it within the periplasm[5–7].

In the absence of a *comEA* orthologue in *H. pylori*, we searched for proteins able to bind the transforming DNA in the periplasm. Here, we showed that ComH is the *H. pylori* periplasmic DNA receptor. The *comH* gene (*hp1527*) is critical for natural transformation, the transformation frequencies of the *comH* null mutants being close to the detection limit[26,36] (Table 1). Amino acid sequence analysis of ComH indicated that it is unique to *H. pylori* with apparently no homology to known protein motifs or domains, making it difficult to speculate on its function during competence[26]. The only recognisable feature of the sequence is a

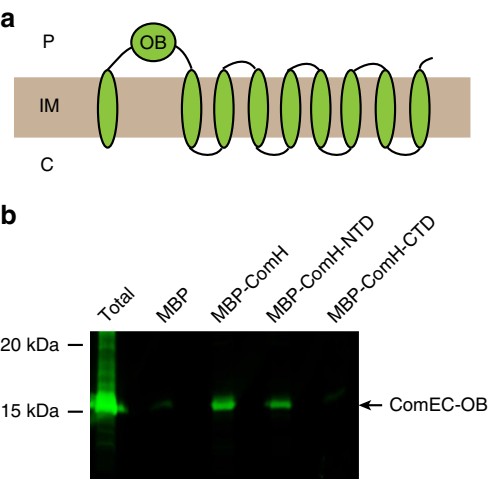

**Fig. 6** ComH interacts with the OB domain of ComEC. **a** Schematic representation of the predicted topology of *H. pylori* ComEC as described in ref. [35]. P = periplasm; C = cytoplasm; IM = inner-membrane; OB = Oligonucleotide binding domain. **b** MBP pull-down assay demonstrating the interaction between *E. coli* expressed ComEC-OB-His₆ and MBP-ComH or MBP-ComH-NTD. The presence of ComEC-OB-His₆ in the fractions eluted from the amylose resin was detected by Western blots using anti-His antibodies.

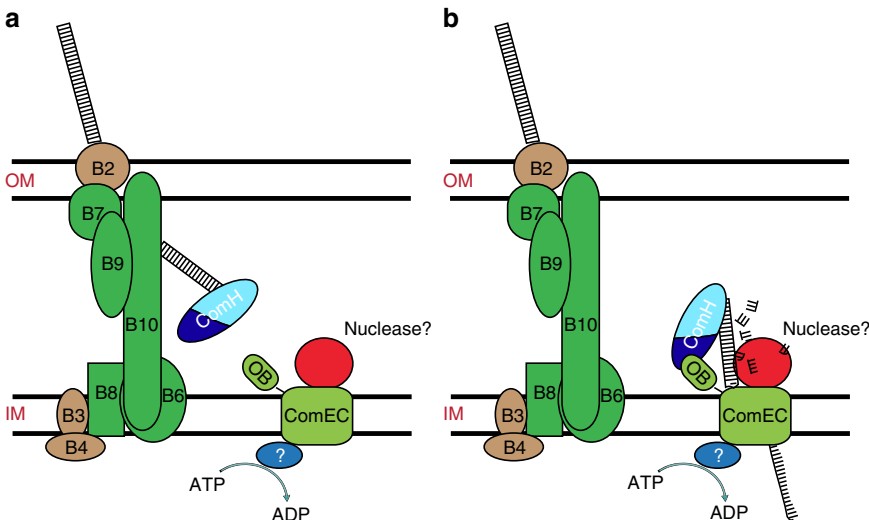

**Fig. 7** Putative roles of ComH in the transport of tDNA through the bacterial envelope. **a** ComH binds to double-stranded DNA through its C-terminal domain (light blue) and allows the retention of the dsDNA into the cytoplasm as proposed for ComEA in *V. cholerae*[5–7]. **b** Through its N-terminal domain (dark blue) interaction with ComEC OB domain, ComH feeds the transforming DNA to the inner-membrane channel that mediates the translocation of the DNA into the cytoplasm as a ssDNA.

putative signal peptide that would target the protein to the periplasm. Despite complete sequence dissimilarity to ComEA proteins, we found that ComH shows the essential characteristics of periplasmic DNA receptor proteins. Indeed, ComH is present in the periplasm, binds dsDNA with high affinity and is required for the formation of tDNA foci, all properties shared with ComEA. While most of the ComEA proteins harbour conserved structural helix-hairpin-helix (HhH) motifs crucial for the DNA interaction[6], ComH has no obvious DNA binding domain. We show that its C-terminal domain is responsible for the dsDNA binding and thus contains a novel DNA binding fold. Further structural studies should allow its definition. As it is the case for ComEA with the Type IV pili involved in the DNA uptake in other species, whether ComH interacts with the ComB components remains to be determined.

How the DNA is handed to the ComEC inner-membrane channel is still unknown. It has been proposed that the presence in the periplasmic space of ComEC loops or domains would allow the interaction of the membrane-embedded channel with other DNA-uptake machinery proteins[13,37]. Interestingly, the capacity of ComH CTD to bind dsDNA is sufficient for the import of the incoming DNA into the periplasm but not for its translocation into the cytoplasm, indicating a separation of functions. Truncation of the NTD leads to the persistence of DNA foci in the periplasm, a phenotype observed in *comEC* mutants[4,31]. These results suggest that this domain is implicated in the delivery of the DNA to the inner membrane pore. Our observation of a physical interaction between ComH NTD and the OB domain from *H. pylori* ComEC, predicted to be exposed to the periplasm, is consistent with a role for ComH in the handing of the DNA to the inner membrane ComEC channel. It would be interesting to investigate if in the case of other naturally transformable bacteria ComEA is also implicated in the delivery of the tDNA to ComEC (ComA in *Neisseria*).

Based on the results presented here, we propose that ComH represents a new class of DNA binding proteins and is not only the missing DNA receptor for natural transformation in *H. pylori* but also the protein responsible for delivering the DNA to ComEC. Its role in the translocation of tDNA through the bacterial envelope can be divided in two steps as represented in Fig. 7. By its capacity for binding dsDNA through its CTD, ComH acts as a receptor for the incoming DNA into the periplasm (Fig. 7a). Subsequently, ComH could deliver the DNA to the inner-membrane channel through the interaction of the NTD with ComEC OB domain (Fig. 7b). Although we have here unveiled the role of a new player in the mechanism of transforming DNA internalisation in *H. pylori*, several questions remain open. In *S. pneumoniae*, the nuclease EndA is required to generate the single stranded DNA capable of crossing the inner-membrane but a protein with similar function remains to be identified in other species[2]. It has been proposed that ComEC could harbour such an activity[35,38]. Furthermore, the need of other yet unknown proteins to process the dsDNA or to coordinate the different steps of the DNA passage through the periplasm cannot be ruled out.

## Methods

***H. pylori* growth conditions**. *H. pylori* cultures were grown under microaerophilic conditions (5% O₂, 10% CO₂, using the MAC-MIC system from AES Chemunex) at 37 °C. Blood agar base medium (BAB) supplemented with 10% defibrillated horse blood (AES) was used for plate cultures. Liquid culture were grown in brain heart infusion media (BHI) supplemented with 10% defibrillated and decomplemented fetal bovine serum (Invitrogen, Carlsbad, CA, USA) with constant shaking (180 rpm). Antibiotic mix containing polymyxin B (0.155 mg/ml), vancomycin (6.25 mg/ml), trimethoprim (3.125 mg/ml), and amphotericin B (1.25 mg/ml) was added to both plate and liquid cultures.

**Construction of gene variants**. *H. pylori* 26695, a kind gift from Agnès Labigne (Institut Pasteur), was used as parental strain to generate all the gene variants listed in Supplementary Table 2. The gene sequences were obtained from the annotated complete genome sequence of *H. pylori* 26695 deposited at Pylori Gene World-Wide Web Server (http://genolist.pasteur.fr/PyloriGene/). For chromosomal integrations/replacements of gene variants, the specific gene regions (with 200–300 bp of flanking sequences wherever appropriate) were amplified from genomic DNA (gDNA) of *H. pylori* 26695 strain using sequence specific primers (Supplementary Table 3) and cloned in pjET1.2 (2974 bp) vector by blunt end ligation. Further, the cassette encoding either the antibiotic resistance or indicated protein tag were introduced in the gene construct by either classical restriction-ligation method or using sequence- and ligation-independent cloning (SLIC). All the plasmids harbouring different gene constructs (listed in Supplementary Table 4) were verified by DNA sequencing and introduced into *H. pylori* by natural transformation. Transformants were selected by using appropriate required antibiotics: kanamycin (20 µg/ml), apramycin (12.5 µg/ml), and chloramphenicol (8 µg/ml). The correct constructions were verified by PCR using locus and gene specific primers. The glycerol stocks of *H. pylori* were prepared in BHI media supplemented with 12.5% glycerol and stored in -80 °C. The details of different constructions generated in this study are given below.

**Construction of *comH::Km/Cm*.** hpcomH (hp1527) locus was amplified using primers Op81 and Op82 and ligated to blunt pjET1.2 vector to generate p1196 (pJet1.2-*comH*).The PCR fragments generated by amplification of p1196 (using primers Op83 and Op84), non-polar kanamycin (using primers KpnI-Km-for and BamH-Km-rev), and chloramphenicol (using primers KpnI-Cm-for and BamHI-Cm-rev) resistance cassette were digested using KpnI and BamHI and ligated separately to generate p1198 (pJet1.2-*comH::Km*) and p1295 (pJet1.2-*comH::Cm*). p1198 and p1295 were used for allelic replacement of native *comH* locus.

**Ectopic expression of ComH-FLAG and ComH-CTD-FLAG.** hp1527 locus (+200 bp upstream sequence) was amplified using Op99 and Op100 and ligated to *rdXA::Cm* cassette present in the plasmid p1081 (amplified using primers Op101 and Op102) using SLIC to generate plasmid p1235 (pJet1.2-Prom-*comH*-FLAG-Cm). The gene sequence coding for ComH-NTD was deleted by reverse PCR of p1235 using Op200 and Op202 to generate plasmid p1337 [pJet1.2-*comH*-CTD (158–479)-FLAG-Cm]. p1235 and p1337 were used to transform *wild-type* strain followed by disruption of the native *comH* locus using p1198. To ectopically express ComH-CTD-FLAG in GFP expressing bacteria, the chloramphenicol cassette of p1337 of was replaced by apramycin cassette (plasmid p1457) and the plasmid was used to transform strain LR887 (26695 pUreA-GFPmut2-Km) followed by disruption of native *comH* using p1295.

**Constructions of translational fusions.** +/- 300 bp at the end of *hp1527* locus was amplified (using primers Op86 and Op104) and cloned in pJet 1.2 vector to generate plasmid p1225. DNA sequence coding for linker-mCherry-Cm (amplified from plasmid p1146 using primers Op107 and Op108) was inserted before the STOP codon of comH gene (p1225 amplified using Op105 and Op106) using SLIC to generate p1236 (pJet1.2- *comH*-linker-Cm). Further, BirA-HA coding sequence was replaced with mCherry of p1236 to generate p1320 (pJet1.2- *comH*-BirA-HA-Cm). These plasmids were then used for allelic replacement of native *comH* locus of the *wild-type* strain.

**Determination of recombination frequencies.** Natural transformation assays were performed as described in[39]. Briefly, exponentially growing *H. pylori* cells (optical density of 4.0 at 600 nm) were incubated with isogenic streptomycin resistance total chromosomal DNA (200 ng) on Blood Agar Base (BAB) plates for 24 h at 37 °C. Next day, serial dilutions of *H. pylori* cells were plated on BAB plates with or without streptomycin (10 µg/ml) and incubated for 4–5 days at 37 °C. The recombination frequencies were calculated as the number of streptomycin resistance colonies per recipient colony-forming unit. P values were calculated using the Mann–Whitney U test on GraphPad Prism software.

**Determination of spontaneous mutation frequencies.** Exponentially growing *H. pylori* cells (optical density of 4.0 at 600 nm) were spotted on Blood Agar Base (BAB) plates for 24 h at 37 °C (no exogenous DNA was added). Next day, the spots were collected in peptone water. 200 µl of non-diluted sample was directly plated on BAB plates with streptomycin (10 µg/ml). In total 20 µl of $10^{-5}$ dilution was plated on BAB plates. The plates were incubated for 6 days at 37 °C. The mutation frequencies were calculated as the number of streptomycin resistance colonies per colony-forming unit.

**Determination of recombination frequencies after electroporation.** Chemically synthesised 139-mer ssDNA (Op245) (Supplementary Table 5) carrying A128G mutation in the *hp1197* gene was used as substrate for electroporation experiment. Electro-competent cells were prepared by re-suspending exponentially growing *H. pylori* cells (optical density of 10 OD/ml at 600 nm) in 1 ml of ice-cold 15% Glycerol, 9% Sucrose. After centrifugation (2204 g) for 3 min), the bacterial pellet was re-suspended in 250 µl of ice-cold 15% Glycerol, 9% Sucrose. 50 µl of these electro-competent cells were mixed with 1 µg of salt free ssDNA in a clean electroporation cuvette. The electroporation was performed at 2.5 kV cm$^{-1}$, and 25 µF. Immediately, 100 µl BHI was added and 50 µl cells were spotted on BAB plates. Next day, serial dilutions of *H. pylori* cells were plated on BAB plates with or without streptomycin (10 µg/ml) and incubated for 4–5 days at 37 °C. The recombination frequencies were calculated as the number of streptomycin resistance colonies per recipient colony-forming unit. P values were calculated using the Mann–Whitney U test on GraphPad Prism software.

**Subcellular fractionation.** To obtain the subcellular fractions, 20 ml of bacterial cultures at $OD_{600}$ 1.5 were pelleted, washed in PBS and resuspended in 2 ml buffer C (20% Sucrose, 1 mM EDTA, 30 mM Tris-HCl, pH8) and further incubated for 10 min at room temperature and centrifuged at $5700 \times g$. The pellet was resuspended in 1 ml of ice-cold water and incubated 10 min at 4 °C. After centrifugation at 5,766 g, the supernatant corresponding to the periplasmic fraction was recovered. The pellet was resuspended in 1 ml of buffer A (10 mM Tris–HCl, pH 7.5, 1 mM DTT, 1× protease inhibitor cocktail) and sonicated. After centrifugation at $147,420 \times g$ the cytoplasmic (supernatant) and membrane (pellet resuspended in 0.2 ml of 30 mM Tris pH 8, 100 mM NaCl) fractions were recovered. The presence of proteins in different bacterial compartments was monitored by immunoblotting.

Uncropped images are provided as Source Data. As controls for the fractionation experiments, we determined the localisation of MotB and NikR, membrane and cytoplasmic proteins, respectively.

**Duplex PCR assay.** The presence of transforming DNA in the periplasm of *H. pylori* was detected using protocol adapted from ref. [6]. Linear plasmid DNA (6.3 kb) was prepared by restriction digestion of p1391 by NcoI-HF. Exponentially growing *H. pylori* cells (1 OD/ml at 600 nm) were incubated with 250 ng of linear plasmid DNA in BHI supplemented with 10% fetal bovine serum for 15 min at 37 °C under microaerophilic conditions. The cells harvested by centrifugation were washed twice with 1× PBS. The excess non-internalised DNA was degraded by nuclease cocktail [DNaseI (50 U/ml), Benzonase (62.5 U/ml) in 1× PBS, and 10 mM $MgCl_2$] treatment at 37 °C for 20 min. The cells were further washed twice, and re-suspended in 200 µl of 1X PBS. This resuspension was subjected to phenol-chloroform extraction and the supernatant was recovered. 1 µl of supernatant was directly used as template for PCR using Q5 DNA polymerase. A 1.1 kb donor specific region was amplified using primers Op284 and Op285, while a (1.4 kb) *H. pylori* gDNA specific region (*hp0247*) was amplified using primers Op31 and Op32. The PCR fragments were visualised by agarose gel electrophoresis (1%) stained with CYBR-Gold (1:10,000 dilution). Uncropped gel images are provided as Source Data.

**Fluorescently labelled DNA substrates for microscopy.** DNA substrates were prepared, as described in ref. [31]. Briefly, 408 bp ATTO-550-dUTP labelled dsDNA was prepared by amplification of *hp1197* locus from 26695 gDNA (100 ng) using primers 1197–5' and 1197–3' (0.5 µM each), 250 µM of dNTP mix, 5 U of ExTaq enzyme (Takara) supplemented with 10 µM of ATTO-550-aminoallyl-dUTP (Jena bioscience). 2 kb ATTO-488-dUTP labelled dsDNA was prepared by amplification of *hp1197* locus from 26695 gDNA (100 ng) using primers OSF342 and OSF343 (0.5 µM each), 100 µM (of dATP, dCTP, dGTP), 50 µM dTTP, 5 U of ExTaq enzyme (Takara), supplemented with 50 µM ATTO-448-aminoallyl-dUTP (Jena bioscience). 100 µl of total reaction volume was maintained and elongation was performed at 72 °C (2 min per kb). The amplified PCR products were purified by Illustra GFX purification kit (GE Healthcare Little Chalfont, UK).

**Fluorescence microscopy experiments.** Both live and fixed *H. pylori* cells were prepared as described[31]. To monitor the fractions of bacteria that formed DNA foci, exponentially growing *H. pylori* cells were incubated with 200 ng of ATTO-550-dUTP labelled dsDNA (408 bp) for 7 min at 37 °C, the unbound DNA was washed and the bacteria were resuspended in BHI and covered with low melting agarose (1.4%) supplemented with 10% fetal bovine serum and imaged immediately. To monitor the time dependent internalisation of ATTO-550-dUTP labelled dsDNA (408 bp) in GFP expressing bacteria, the samples were prepared as described above and the bacteria that formed DNA foci were observed under live conditions [gas mixture (10% CO$_2$, 3%O$_2$), humidity (90%)] at 37 °C for 3 h. To monitor *in vivo* co-localisation of ComH-mCherry with DNA, 1 µg of ATTO-488-dUTP labelled dsDNA (2 kb) was incubated with exponentially growing *H. pylori* cells for 90 min at 37 °C under microaerophilic conditions. The bacteria harvested by centrifugation (323 g for 3 min) were subjected to DNaseI (50 U/ml) treatment at 37 °C for 5 min followed by 1× PBS wash before fixation with formaldehyde (4%). All the images were captured with a 60x objective using inverted Nikon A1R confocal laser scanning microscope system equipped with an environmental chamber. The images were processed and analysed using NIS-element software (Nikon Corp., Tokyo, Japan) and ImageJ software. The bacteria that formed DNA and ComH-mCherry foci were counted manually. The volumes of internalised DNA were calculated by 3-D image analysis performed using Volocity software (Perkin Elmer, Waltham, USA).

**Cloning, over-expression and purification of ComH and constructs.** *Escherichia coli* strains used for cloning (DH5α), and for protein overexpression, and purification (BL21) were cultured in Luria–Bertani (LB) broth, or LB agar plates supplemented with the required antibiotics [ampicillin (100 µg/ml), kanamycin (50 µg/ml), apramycin (50 µg/ml), and chloramphenicol (34 µg/ml)]. To purify the C-terminally His$_6$ tagged proteins under the control of IPTG inducible promoter, the sequence encoding amino acids 20–479 of ComH (amplified using Op220 and Op221) was ligated to pnEA-vH (amplified using primers Op219 and Op226) using SLIC to generate plasmid p1364 [pnEA-vH –*comH (20–479)-His$_6$*]. His$_6$ tagged fusion constructs of ComH-NTD (amino acids 20–169) and ComH-CTD (amino acids 170–479) were generated by reverse PCR of p1364 using primer pairs (Op239 and Op240), and (Op243 and Op244) respectively. To purify the MBP fused constructs under the control of IPTG inducible promoter, the sequence encoding amino acids 20–479 of ComH (amplified using Op127 and Op82) was ligated to pMAL-p2X (amplified using primers Op128 and Op129) using SLIC to generate plasmid p1249 [pMAL-p2X –*MBP-comH (20–479)*]. MBP tagged fusion constructs of ComH-NTD (amino acids 20–169) and ComH-CTD (amino acids 170–479) were generated by reverse PCR of p1249 using primer pairs (Op198 and Op205), and (Op206 and Op201) respectively. All the plasmids containing fusion constructs were verified by DNA sequencing and are listed in Supplementary Table 4.

For large-scale purification, the plasmids carrying different recombinant DNA constructs were transformed into *E. coli* BL21 expression strain and grown in LB media containing ampicillin (100 µg/ml) at 37 °C. Induction of the proteins were achieved by IPTG (0.5 mM) at an optical density of 0.6 at 600 nm for 4–5 h at 30 °C. The cells harvested by centrifugation were resuspended in lysis buffer (50 mM Tris pH 8.0, 500 mM NaCl, 1 mM DTT, 0.1 mM EDTA, 0.2% NP40, and 1× protease inhibitor cocktail). 5 mM imidazole was added in case of the His$_6$-tagged proteins. After sonication, the soluble fractions were separated from the cell debris by centrifugation at $18,637 \times g$ for 45 min. Standard Ni$_2$+-NTA, and Amylose affinity chromatography was used for the first step of purification of proteins. The fractions containing the proteins were pooled, and desalted using HiTrap Desalting Column (GE Healthcare). Full-length ComH, and ComH-NTD were further purified using Resource-S, while the ComH-CTD was further purified using Resource-Q ion-exchange chromatography. Purity of the proteins was analysed by SDS-PAGE and the pure fractions were pooled and dialysed twice for 4–6 h against storage buffer (50 mM Tris, 100 mM NaCl, 1 mM DTT, 0.1 mM EDTA, and 10% glycerol). The proteins were quantitated using Bradford assay and small aliquots were frozen immediately in liquid N$_2$, and stored at −80 °C.

**Electrophoretic mobility shift assay.** HPLC purified, chemically synthesised oligonucleotides (5′-Cy5 or 3′-TAMRA labelled) (Eurogentec) were used for the EMSA experiments (Supplementary Table 5). Different DNA substrates were prepared by annealing appropriate combinations of oligonucleotides (Supplementary Table 5). Indicated concentrations of purified proteins were incubated with DNA substrates (30 nM) in binding buffer (10 mM Tris–HCl pH 7.5, 50 mM KCl, 0.1 mM MgCl$_2$, 1 mM DTT, 0.1 µg/µl BSA). The NaCl concentration was maintained at constant amount (~100 mM) by adding protein storage buffer to the reaction mixture. After incubation of 30 min on ice, the reaction mix was separated by native-PAGE (6%). The gels were visualised by using Typhoon and quantified using ImageJ. Percentage binding was estimated by measuring the depletion in substrate DNA considering DNA without protein as 100%.

**Surface plasmon resonance.** SPR binding kinetics were analysed using the BIAcore3000 optical biosensor (GE Healthcare) at 25 °C. A 50-mer 5′-biotinylated DNA (Op249) was immobilised on a streptavidin-coated static biosensor chip. The double-stranded was generated by hybridising the complementary DNA strand (Op250). Variable concentrations of ComH-His6 (0–5 µM) in binding buffer (PBS, and 0.005% Tween20) were injected into the biosensor chip for 240 s followed by dissociation time of 900 s.

**Microscale thermophoresis.** DNA binding kinetics using microscale thermophoresis were carried out on a NanoTemper´s Monolith NT.115 instruments. Briefly, an 18-mer 5′-fluorescently labelled oligonucleotide with a FAM probe was hybridised with the complementary DNA. Increasing concentrations of ComH-His$_6$, ComH-CTD-His$_6$ and ComH-NTD-His$_6$ were used (typically 0.3 nM to 5 µM) in buffer 50 mM Tris-HCl pH 7.4, 150 mM NaCl, 10 mM MgCl$_2$, 0.05% v/v Tween20. The signals to noise ratio are above 12 as recommended (signal is defined as the response amplitude (difference of Fnorm between DNA alone and DNA-protein complex) and the noise, defined as the standard deviation of errors between replicates). They range between 26 to 42 for three interactions. The difference of Fnorm is positive for ComH and ComH-NTD and negative for ComH-CTD as already observed for other projects.

**Sedimentation velocity analytical ultracentrifugation (SV-AUC).** The interactions between ComH-His6 and dsDNA or ssDNA were measured using an Optima proteome-lab XLI (Beckman Coulter, Palo Alto, USA) with an An-50 Ti rotor at 20 °C equipped with a fluorescent detector (Aviv) by doing an isoterm analysis, a titration of dsDNA by ComH-His6. 400 µL of samples were centrifuged at $128,297 \times g$. Sedimentation profiles were collected every 5 min and analysed as described[40–42]. A 18-mer 5′-fluorescently labelled oligonucleotide with a FAM probe was used as ssDNA. The 18 bp dsDNA was formed by hybridising the 18nt ssDNA with the complementary DNA. Increasing concentrations of ComH-His6 (between 3 nM to 2 µM) were added to 60 nM of fluorescently labelled oligonucleotids in buffer Tris-HCl 50 mM pH 7.4, NaCl 150 mM, MgCl2 10 mM, Tween 20 0.05% v/v. We observed a transient population with a sedimentation coefficient of 3.8S and a high sedimentation complex with sedimentation coefficient of 6.5S. The $K_D$ for 3.5S and 6.5S complexes are 0.1 and 0.3 µM and may correspond to ComH-dsDNA and [ComH]$_2$-dsDNA, respectively. A 18-mer 5′-fluorescently labelled oligonucleotide with a FAM probe was used as ssDNA. The 18 bp dsDNA was formed by hybridising the 18nt ssDNA with the complementary DNA. Increasing concentrations of ComH-His6 (between 3 nM to 2 µM) were added to 60 nM of fluorescently labelled oligonucleotids in buffer 50 mM Tris–HCl pH 7.4, 150 mM NaCl, 10 mM MgCl$_2$, 0.05% v/v Tween 20. We observed a transient population with a sedimentation coefficient of 3.8S and a high sedimentation complex with sedimentation coefficient of 6.5S. The $K_D$ for 3.5S and 6.5S complexes are 0.1 and 0.3 µM and may correspond to ComH-dsDNA and [ComH]$_2$-dsDNA, respectively.

**MBP pull-down assay.** The coding sequence (amino acids 30–156) of *hpcomEC* amplified using primers HpComEC-30_F and HpComEC-156_R2 was ligated to plasmid pnEAvH (digested using NdeI + BamHI) using SLIC to generate plasmid p1397. The interaction between MBP-ComH and ComEC-OB-His$_6$ was tested by amylose affinity pull-down assay. The plasmids carrying these recombinant constructs were transformed in *E. coli* BL21 cells. The expression of the proteins was induced by IPTG (0.5 mM) and cultures were grown for 3 h at 37 °C. Equal amounts of these *E. coli* cultures were mixed together and sonicated. After centrifugation, the soluble fraction of the cell lysate was incubated with amylose resin for overnight at 4 °C to allow the interaction between the proteins. The resin was washed, and the bound proteins were eluted using maltose (10 mM). Equal amounts of the elution fractions were separated by SDS-PAGE. The presence of ComEC-OB-His$_6$ in the eluted fractions was detected by Western blots using anti-His antibodies. Uncropped gel images are provided as Source Data.

**Reporting summary.** Further information on research design is available in the Nature Research Reporting Summary linked to this article.

## Data availability
The authors declare that the main data supporting the findings of this study are available within the article and its Supplementary information files. The source data underlying Figs. 1b-d, 2a, 3a, c, 4a-b, d are provided as Source Data. Extra data are available from the corresponding author upon request.

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

## Acknowledgements

We thank Didier Busso for providing the plasmids used for cloning. We thank IRCM Microscopy and Molecular Biology (CIGEX) facilities for technical assistance and Anna Campalans for her help with cell imaging. We thank Audrey Comte for her help with biophysical measurements and Hilde De Reuse (Institut Pasteur) for antibodies against MotB and NikR. Financial support for this work was provided by the CEA, INSERM and grants from the Indo-French Centre for Promotion of Advanced Research (CEFIPRA grant n° 5203–5). DNR acknowledges DST for JC Bose Fellowship. Biophysical measurements were performed on I2BC platform PIM that is supported by the French Infrastructure for Integrated Structural Biology (FRISBI) ANR-10-INBS-05. P.P.D. was supported by grants from *Enhanced Eurotalents* fellowship programme (CEA/EU), CEFIPRA and the Région Ile de France (DIM1Health).

## Author contributions

P.P.D., A. M. D. G., J.B.C., and J.P.R. designed the experiments. P.P.D., A.M.D.G., S.M., C.V., P.F.V., X.V., J.P.R., M.M. and G.V.G. performed the experiments. P.P.D., A.M.D.G., J.B.C. and J.P.R. analysed the data. P.P.D. and J.P.R. wrote the paper. D.N.R. and J.P.R. acquired funding. All authors provided critical inputs and approved the final version of the manuscript.

## Competing interests

The authors declare no competing interests.
