## [Peer Review File · Nature Communications]

Reviewers' comments:

Reviewer #1 (Remarks to the Author):

The manuscript „Identification of the periplasmic DNA receptor for natural transformation of *Helicobacter pylori*“ delivers novel results on the ComH protein, which is unique to the bacterium. The topic is highly relevant, since genetic exchange in this human pathogen drives co-evolution and adaptation to the human host. The data are extremely interesting for a broad readership and will lead to more understanding of the DNA uptake process in bacteria in general. For me it was a pleasure to read this manuscript. In particular, the data support the conclusion that ComH is a novel periplasmic DNA-binding protein implicated in natural transformation in *H. pylori*. Furthermore, the C-terminus of ComH was shown to bind dsDNA (with only low affinity for ssDNA) and that this C-terminal domain is sufficient to complement a comH deletion mutant for DNA transport across the outer membrane but not across the inner membrane. Besides, the authors provide evidence that ComH interacts with ComEC, the general inner membrane channel, used in all so far known competent bacteria, via the N-terminal domain. This suggests a further role of ComH in transfer of DNA to the ComEC transporter. These results are novel and per se very important for the understanding of the mechanism of uptake of huge amounts of DNA observed for this pathogen.

Solid biochemical evidence is delivered for the above mentioned features of ComH. However, the interpretation that ComH might have a similar role than the DNA-binding protein ComE in other organisms, meaning that binding of DNA by ComH pulls the macromolecule across the outer membrane, is not supported by the data and actually also not expected considering the kinetics of *H. pylori* DNA uptake. At external forces of 10 pN, the velocity of the ComB system vs. the ComE DNA-binding dependent DNA transport system is at least tenfold faster (around 0.5 $\mu\text{m/s}$ in Hp vs. 0.03 $\mu\text{m/s}$ in Ng, Stingl et al. 2010 and Hepp & Maier 2017). Besides, the amount of imported DNA is limited to around 40 kbp using the DNA-binding protein ComE (Gangel et al. 2014), while it seems unlimited in *H. pylori*, capable of transport of 1.6 Mb within 10 min (Krueger et al. 2016). This notion should be addressed in more detail in the manuscript (see below).

Points to be answered:

1. How is the distribution of ComH-mCherry without contact to DNA?
2. How was the co-localization of ComH-mCherry and ATTO-488-DNA quantified (see Fig. 2)? The authors mention that this was done “manually” (page 15, 2nd last sentence); a qualitative analysis is not sufficient, since ComH-mCherry was detected throughout the bacterium; thus, a quantitative threshold has to be defined for the definition of a ComH-mCherry focus, which might co-localize with the DNA focus. I suggest to quantitatively reanalyze the fluorescence intensities of given ROIs in both channels (for mCherry and ATTO-488) and correlate these quantitative results with each other. Does ComH amount quantitatively correlate with the amount of imported DNA? In addition, the authors should also quantify the mCherry foci in the cells without DNA in order to evaluate the impact of DNA import as trigger (or not) for focal distribution of ComH in the periplasm (see also point 1.). Did the authors also observe DNA foci with no co-localized ComH-mCherry? These results will substantially add information about the role of ComH.
3. It would be important to see the co-localization of ComH with DNA in highly competent cells. In this manuscript, only *H. pylori* with 20 % DNA import was analyzed (page 4, 2nd chapter). What happens if competence development and increase in DNA uptake per cell is triggered to 60-80 % competent cells with more than one DNA focus per cell by e. g. aerobic incubation for 1-2 h and/or increase in pH?
4. An alternative role of ComH might be the activation of the ComB transport complex and a restricted transfer of incoming DNA to the ComEC complex; a huge amount of DNA is taken up

over the outer membrane with high velocity, which is not comparable to the low velocity uptake via binding to ComE in other organisms (see above); it would be intriguing, if the role of ComH is to prevent extensive uptake of the vast amount of periplasmic DNA into the cytoplasm. If this hypothesis is true, the authors should see less quantitatively co-localized ComH with increasing amounts of DNA (see point 4.).

5. Why did the authors incubate the cells in the presence of DNA for 90 min for co-localization experiments but only for 7 min to monitor the fraction of bacteria with DNA foci? What was the fraction of competent cells compared to Wt (in Suppl Tab. 1 natural transformation was only marginally affected with 60 % of Wt level). If competence development was substantially reduced, the experiments have to be compared with experiments performed under conditions suggested in point 4. . If competence development was similar to Wt, how was the co-localization after 7 min? Does ComH co-localize with imported DNA in a time-dependent manner?

6. In the manuscript various fusion ComH proteins were constructed and detected (ComH-mCherry, ComH-His6, ComH-MBP, ComH-FLAG, ComH-BirA-HA); the stability of the protein and the specificity for detection is missing for ComH-mCherry and should be provided by showing Western Blots (e. g. in Fig. S1 in analogy to MBP and His6). Furthermore, the authors should comment on why they chose which fusion protein for each experiment.

7. Fig. 2a: Why do the authors present the ComH-BirA-HA fusion here? As shown in Supp. Tab. 1, this fusion has only 2 % of the Wt function. Also, the protein bands on the gel could be clearer.

8. Delete Fig. 6a and the movies 1-3: The data do not add any information beyond the results shown in Fig. 5 (the latter results from natural transformation, electroporation, PCR from transferred and DNase protected fragments and fluorescent DNA uptake into the periplasm) are sufficient for demonstrating that ComH-CTD can complement DcomH for transport across the outer membrane but not for transport across the inner membrane). It is not obvious that ATTO-488 is really transported into the cytoplasm. A resolution of 100 nm per pixel with a periplasmic space of ~30 nm is unsuitable to measure location of DNA. In addition, the authors showed previously, that ATTO-488 labelled DNA harboring a GFP marker did not lead to GFP expression inside the cells (while unlabeled DNA did lead to 50 % of expression) and one plausible reason might be that the covalently labelled DNA is excluded from the cytoplasm. Therefore, in the previous paper the abstract reads that labelled DNA was "eventually internalized into the cytoplasm". Can the authors exclude some DNase activity in the periplasm, leading to slow (!) degradation of DNA after long-term exposure (like shown in the movies). Transport into the cytoplasm was observed to be much faster (completed within 30-60 min) than the long-term experiment shown by the authors (up to 3.5 hours!). Actually, by eye there is even no difference in DNA foci stability visible between Wt and the comEC mutant in the selected movies.

Hence, the results provided in Fig. 5 are solid and show clearly the essential function of ComH and the C-terminal domain in transport of DNA across the outer membrane and the need for the N-terminal domain for full function of outer and inner membrane DNA transport.

9. Method section: Information about NikR and MotB-tagging is missing. For ease of reading, please sort the Method section starting with all genetic constructions, separated in *H. pylori* and *E. coli*, biochemical assays and bacterial assays. Also information about the Western blots, including the used antibodies is missing.

I highly recommend this manuscript for publication if the points above have been addressed by the authors. The results will substantially contribute to higher understanding of this extremely important topic of genetic exchange in *H. pylori*.

Reviewer #2 (Remarks to the Author):

Damke et al describe that ComH, which shows a periplasmic location, is essential for the import of dsDNA into the periplasm, but it is not involved in the internalization of ssDNA into the cytoplasm. The manuscript resolves many of the puzzles of the DNA uptake machinery of *H. pylori* competent cells, but before publication the authors must solve the problem with the DNA binding assays.

General comments

1. Page (P3), lane (L) 6-7. I recommend not to quote the statement that "H. pylori imports up to 1.6 Mb of transforming DNA into the periplasm" until any other independent confirmation is reported. I have some problems with the numbers reported in the original paper. Since the H. pylori type-IV secretion system imports dsDNA into the periplasmic space at a velocity of 1,300-bp/s, and it is expected that ComEC transport ssDNA through the inner membrane at a velocity of ~80-bp/s, we have to assume that dsDNA has to accumulate into the periplasm. H. pylori is 3 μm long with a diameter of about 0.5 μm and an expected total volume of the periplasm of $\sim 4 \times 10^{-16}$ L. Perhaps the information is correct, but to accept that up to 1.6 Mb of linear dsDNA (1.6 Mb) of tDNA can accumulate into the busy periplasm an independent confirmation is required.

2. P3, L 12 from bottom. The sentence is misleading, absence of ComH reduced DNA uptake rather than the recombination frequency. Inactivation of comEC leads to $\sim 2 \times 10^{-9}$ transformants or spontaneous mutants in its target sequence. I wonder what is the spontaneous mutation rate in specific H. pylori strains? I thought that mutations are on average more frequent than recombination events on nature.

3. P4, L 13-14. In the absence of comH ~1% of cells accumulate fluorescently labelled dsDNA or ~20% of total dsDNA (donor/recipient rate) is accumulated in the periplasmic space, but the transformation frequency drops >400,000-fold. Please elaborate on such discrepancies.

4. P5, L11-12 from bottom). From the data presented in Fig. 2b, there is a discrepancy in the ratio of colocalization of DNA (4 green dots, one faint) with ComH (8 red dots, and >6 faint dots). I can see that 100% of the tDNA foci colocalized with the ones of ComH.

5. P6, L8. Are competent comH-His variants proficient in natural transformation? The microscale thermophoresis data revealed a very high dispersion rate (Fig. 3c), I wonder how the authors reach a Kd value.

6. P6, L11-12. The authors assumption "The fusion protein MBP-ComH exhibits a slightly higher affinity compared to ComH-His6 probably due to a stabilizing effect of the MBP protein" jeopardized all DNA binding assays. Alternatively, the presence of His-tag misfolds the ComH protein, and ComH-His is a mutant variant. There are many papers describing this, even rendering fused proteins more thermosensitive. MBP-ComH binds ssDNA with lower affinity compared to dsDNA, but ComH-His does not bind ssDNA.

7. P7, L14-15. The authors stated that "The Kd obtained for ComH-CTD (2.9 μM) (Fig. 4d) was comparable to that obtained for the full-length protein (1.25 μM) (Fig. 3c). However, from the data show in Fig. 4c we can deduce a Kd of ~ 0.4 μM for ComH-CTD-His and of ~ 0.75 μM for full-length ComH-His. Furthermore, it can be deduced a Kd of ~ 0.1 μM for both MBP-ComH-CTD and full-length MBP-ComH (Fig. S4b). Please remove the tag or focus in the MBP-ComH variants.

8. P7, L14. With the protein concentration used, the authors cannot reach a conclusion about the substrate preference.

Specific comments

P2, L16. Please correct. The authors stated that ComE(A) is conserved amongst most naturally transformable Gram-positive and Gram-negative bacteria, but, although conserved, in some bacteria ComEA is membrane bound, whereas in others ComE or ComE(A) is free in the cytoplasm.

Reviewer #3 (Remarks to the Author):

When it comes to natural competence for transformation, the human pathogen *H. pylori* is particular and different from most other studied competent Gram-negatives, as it uses a type IV secretion system instead of a type IV pilus as central part of its DNA-uptake machinery. As such, an open question in the field has been as to which protein would accept the incoming genetic material in the periplasm of this organism, given that a homolog of the DNA-binding protein ComE/ComEA is missing. In this study, Damke et al characterized a recently identified competence protein (Ref#26), ComH, and provide evidence that this protein is responsible for DNA import into the periplasm. In addition the protein interacts with the inner membrane DNA translocator ComEC though this interaction has not been addressed on the mechanistic level.

The study is well performed and combines bacterial genetics, basic biochemistry, and imaging. Indeed, the authors used well thought-through experimental designs such as the ssDNA electroporation assay or a PCR-based DNA uptake assay to provide convincing data that helped identifying the step in which ComH acts in the DNA uptake process. They further investigated this finding - mostly through in vitro assays.

The first part of the paper (KO, complementation, mCherry fusion, DNA co-location, purification and EMSA) is well performed and very clear but somewhat recapitulates previous studies on ComEA (e.g., ref#6). The second part of the paper is novel in a way that it identified separate regions of the ~55kDa ComH protein (much larger than the known small ComEA proteins of 10-15kDa of other Gram-negative bacteria) and assigns these regions to specific functions. Precisely, the C-terminal domain was shown to bind to DNA while the N-terminal domain was shown to bind to the inner membrane DNA translocation channel ComEC.

Overall, this is a conclusive study that will be of importance for the natural transformation community and probably also beyond.

Below I provide suggestions on how to further improve the manuscript.

Major points:

- While the interaction of ComH with dsDNA is shown with 3-4 different methods, the interaction with ComEC would benefit from an additional approach to support the data (e.g., pull-down from native cell lysates, BTH or similar approaches). Indeed, with the current approach both proteins are synthesized in *E. coli* and then mixed. But how would this look like in the periplasm?
- At this stage, it is also theoretically possible that the purified ComH already binds DNA from the *E. coli* cytoplasm (as was the case for ComEA; ref#6) and that the OB-binding domain of ComEC binds to the same DNA and not directly to ComH. Also the Histag might bind to the DNA (as for example speculated by Balaban et al 2014, who showed unspecific DNA binding through His-tagged proteins as a result of electrostatic charges/interactions). This should be addressed (e.g., by adding nucleases to the pull-down assay).
- Fig. 2B: The authors wrote "Two ComH-mCherry distribution patterns were observed. While the majority of the cells showed a rather homogenous distribution of the protein, in about a third of the cells it formed distinct clusters (Fig. 2b)." This is not really reflected in the images. The protein doesn't seem to localize homogeneously but in distinctive regularly interspaced puncta. There is also no obvious periplasmic localization. This inconsistency should be addressed by the authors.
- The discussion seems more a summary of the results than a global picture. I especially miss the upstream part => how do the authors see the interaction of the ComH protein with the T4SS? In

this context, the manuscript would be significantly strengthened if the authors would experimentally address this question and visualize the localization of ComH in the presence and absence of the T4SS and check if there is any co-localisation with membrane-proteins of the T4SS.

Minor points:

- Line numbers would have significantly facilitated the review process.
- Please describe genetic constructs (e.g., page 5: Δ comH rdxA::comH-FLAG => why in rdxA gene? Neutral locus?)
- Page 4: can the authors speculate on the residual 1% of periplasmic tDNA foci in the periplasm?
- The authors should ensure that references are properly used (e.g., citing the references that fit with the organism described in the preceding sentence).
- Page 9 "In *N. gonorrhoeae* and *V. cholerae* it is proposed that the incoming DNA is pulled across the outer membrane by the periplasmic DNA binding protein ComEA (ComE in *Neisseria*) => needs references (Ref#5 to 7).
- Page 10: "The lack of sequence specificity in the binding of ComH to dsDNA is consistent with the previous observation that *H. pylori* displays no sequence bias for DNA uptake." => this sentence is misleading as it implies that ComEA/ComE would exert such a sequence-specificity in those Gram-negatives that distinguish DNA and required DNA uptake sequences (DUS/USS) for import. However, this is not the case, as DNA sorting occurs outside the cell (for example through ComP in *Neisseria meningitidis*; Pelicic group).
- As the authors provide a quantification of panel c in panel d of Fig.1, the same should be done for Fig. 5d.
- Legend figure 7 needs reference.

Point by point answer to comments

Major changes in the text are indicated in red.

Reviewer 1

The manuscript „Identification of the periplasmic DNA receptor for natural transformation of Helicobacter pylori” delivers novel results on the ComH protein, which is unique to the bacterium. The topic is highly relevant, since genetic exchange in this human pathogen drives co-evolution and adaptation to the human host. The data are extremely interesting for a broad readership and will lead to more understanding of the DNA uptake process in bacteria in general. For me it was a pleasure to read this manuscript. In particular, the data support the conclusion that ComH is a novel periplasmic DNA-binding protein implicated in natural transformation in H. pylori. Furthermore, the C-terminus of ComH was shown to bind dsDNA (with only low affinity for ssDNA) and that this C-terminal domain is sufficient to complement a comH deletion mutant for DNA transport across the outer membrane but not across the inner membrane. Besides, the authors provide evidence that ComH interacts with ComEC, the general inner membrane channel, used in all so far known competent bacteria, via the N-terminal domain. This suggests a further role of ComH in transfer of DNA to the ComEC transporter. These results are novel and per se very important for the understanding of the mechanism of uptake of huge amounts of DNA observed for this pathogen. Solid biochemical evidence is delivered for the above mentioned features of ComH.

We thank this reviewer for the positive evaluation of our results.

However, the interpretation that ComH might have a similar role than the DNA-binding protein ComE in other organisms, meaning that binding of DNA by ComH pulls the macromolecule across the outer membrane, is not supported by the data and actually also not expected considering the kinetics of H. pylori DNA uptake. At external forces of 10 pN, the velocity of the ComB system vs. the ComE DNA-binding dependent DNA transport system is at least tenfold faster (around 0.5 $\mu\text{m/s}$ in Hp vs. 0.03 $\mu\text{m/s}$ in Ng, Stingl et al. 2010 and Hepp & Maier 2017). Besides, the amount of imported DNA is limited to around 40 kbp using the DNA-binding protein ComE (Gangel et al. 2014), while it seems unlimited in H. pylori, capable of transport of 1.6 Mb within 10 min (Krueger et al. 2016). This notion should be addressed in more detail in the manuscript (see below).

We agree with this reviewer on the fact that the data we present does not provide evidence for a role for ComH in “pulling” the DNA into the periplasm, although we don’t think it can be completely ruled out, since it was shown that binding of proteins can constitute a driving force in the translocation of DNA by providing a “passive ratchet” by retaining the translocated DNA (Salman H, et al. (2001) Kinetics and mechanism of DNA uptake into the cell nucleus. PNAS U S A 98: 7247–7252). We have therefore changed the text in the abstract and the discussion as well as the model on Figure 7.

1. *How is the distribution of ComH-mCherry without contact to DNA?*

We cannot really answer that question since we have no way to rule out the presence of DNA. Indeed, in normal culture conditions natural transformation occurs spontaneously as revealed by the fact that genetic markers can be transferred from one strain to another by simply co-culturing them, suggesting that chromosomal DNA, possibly from lysed bacteria, is always present in the medium. This could result in the same mixed patterns we observe with or without added DNA. It would also explain the presence of ComH clusters without visible DNA foci. We have now added this explanation to the results section.

- 2. How was the co-localization of ComH-mCherry and ATTO-488-DNA quantified (see Fig. 2)? The authors mention that this was done "manually" (page 15, 2nd last sentence); a qualitative analysis is not sufficient, since ComH-mCherry was detected throughout the bacterium; thus, a quantitative threshold has to be defined for the definition of a ComH-mCherry focus, which might co-localize with the DNA focus. I suggest to quantitatively reanalyze the fluorescence intensities of given ROIs in both channels (for mCherry and ATTO-488) and correlate these quantitative results with each other. Does ComH amount quantitatively correlate with the amount of imported DNA? In addition, the authors should also quantify the mCherry foci in the cells without DNA in order to evaluate the impact of DNA import as trigger (or not) for focal distribution of ComH in the periplasm (see also point 1.). Did the authors also observe DNA foci with no co-localized ComH-mCherry? These results will substantially add information about the role of ComH.*

As suggested by the reviewer, we have now used the ImageJ JACOP plugin to analyse the images and calculate the Manders' correlation coefficient reflecting the overlap of the green (DNA foci) signal with the ComH-Cherry clusters using a threshold for the red fluorescent signal. As explained in 1., we cannot really assess the impact of the absence of DNA in our experiments. We now have modified the text and added the new data in the results section.

- 3. It would be important to see the co-localization of ComH with DNA in highly competent cells. In this manuscript, only H. pylori with 20 % DNA import was analyzed (page 4, 2nd chapter). What happens if competence development and increase in DNA uptake per cell is triggered to 60-80 % competent cells with more than one DNA focus per cell by e. g. aerobic incubation for 1-2 h and/or increase in pH?*

We agree with the reviewer on the idea that analysing the response of ComH localisation to environmental stresses modulating the competence capacity is an interesting point. However, we think that these questions are beyond the scope of this work. Furthermore, because of the reasons mentioned in the precedent paragraphs, the experiments required to answer them are not obvious.

- 4. An alternative role of ComH might be the activation of the ComB transport complex and a restricted transfer of incoming DNA to the ComEC complex; a huge amount of DNA is taken up over the outer membrane with high velocity, which is not comparable to the low velocity uptake via binding to ComE in other organisms (see above); it would be intriguing, if the role of ComH is to prevent extensive uptake of the vast amount of periplasmic DNA into the cytoplasm. If this hypothesis is true, the authors*

should see less quantitatively co-localized ComH with increasing amounts of DNA (see point 4.).

As mentioned above and in the text, even in our experimental conditions we do see many ComH-Cherry clusters which are not associated with labelled DNA foci. This makes the results from the kind of experiments proposed hard to interpret.

- 5. Why did the authors incubate the cells in the presence of DNA for 90 min for co-localization experiments but only for 7 min to monitor the fraction of bacteria with DNA foci? What was the fraction of competent cells compared to Wt (in Suppl Tab. 1 natural transformation was only marginally affected with 60 % of Wt level). If competence development was substantially reduced, the experiments have to be compared with experiments performed under conditions suggested in point 4. . If competence development was similar to Wt, how was the co-localization after 7 min? Does ComH co-localize with imported DNA in a time-dependent manner?*

The choice of 90 minutes for the co-localisation experiments was due to the fact that at that time we saw the maximum frequency of cells with ComH-Cherry clusters. At shorter times very few cells (< 5%) had distinguishable ComH-Cherry clusters. For that reason, the experiments were performed in a *comEC* background, allowing the stabilisation of the DNA foci (Stingl et al., 2010; Corbinais et al., 2016). To monitor the fraction of bacteria with DNA foci, we used shorter times as those were shown to reflect the level of competence (Kruger et al., 2016; Corbinais et al., 2017).

- 6. In the manuscript various fusion ComH proteins were constructed and detected (ComH-mCherry, ComH-His6, ComH-MBP, ComH-FLAG, ComH-BirA-HA); the stability of the protein and the specificity for detection is missing for ComH-mCherry and should be provided by showing Western Blots (e. g. in Fig. S1 in analogy to MBP and His6). Furthermore, the authors should comment on why they chose which fusion protein for each experiment.*

In the case of the MBP and His₆ fusions, the proteins were expressed in *E. coli* and purified to be use in the *in vitro* experiments (DNA binding and interactions). The FLAG and Cherry fusion constructs were designed for expression in *H. pylori*. The new Suppl. Figure 2 shows the expression of the ComH-Cherry fusion protein in *H. pylori* extracts using an antibody against the fluorescent protein.

- 7. Fig. 2a: Why do the authors present the ComH-BirA-HA fusion here? As shown in Suppl. Tab. 1, this fusion has only 2 % of the Wt function. Also, the protein bands on the gel could be clearer.*

We have now confirmed the results of the periplasmic localisation of ComH using the FLAG fusion, which adds just a few amino acids to ComH and yields transformation frequencies similar to those of the wild-type strain (Table 1). The new Fig. 2a shows this result.

- 8. Delete Fig. 6a and the movies 1-3: The data do not add any information beyond the results shown in Fig. 5 (the latter results from natural transformation,*

electroporation, PCR from transferred and DNase protected fragments and fluorescent DNA uptake into the periplasm) are sufficient for demonstrating that ComH-CTD can complement DcomH for transport across the outer membrane but not for transport across the inner membrane). It is not obvious that ATTO-488 is really transported into the cytoplasm. A resolution of 100 nm per pixel with a periplasmic space of ~30 nm is unsuitable to measure location of DNA. In addition, the authors showed previously, that ATTO-488 labelled DNA harboring a GFP marker did not lead to GFP expression inside the cells (while unlabeled DNA did lead to 50 % of expression) and one plausible reason might be that the covalently labelled DNA is excluded from the cytoplasm. Therefore, in the previous paper the abstract reads that labelled DNA was “eventually internalized into the cytoplasm”. Can the authors exclude some DNase activity in the periplasm, leading to slow (!) degradation of DNA after long-term exposure (like shown in the movies). Transport into the cytoplasm was observed to be much faster (completed within 30-60 min) than the long-term experiment shown by the authors (up to 3.5 hours!). Actually, by eye there is even no difference in DNA foci stability visible between Wt and the comEC mutant in the selected movies. Hence, the results provided in Fig. 5 are solid and show clearly the essential function of ComH and the C-terminal domain in transport of DNA across the outer membrane and the need for the N-terminal domain for full function of outer and inner membrane DNA transport.

In our previous work we did indeed find that ATTO488-labelled DNA probably failed to be incorporated into the genome. Our data, in particular the behaviour of the foci in a *comEC* mutant, were consistent with this being a limitation in the processing of the labelled DNA downstream of its entrance into the cytoplasm. If the disappearance of the foci with time was due only to a non-specific degradation of the DNA, we wouldn't expect it to be affected by disabling ComEC. As for the localisation of the DNA foci in the periplasm, we completely agree with the reviewer's comment regarding the lack of resolution to prove it just by microscopy. As we mentioned in our previous work, we inferred that localisation of the foci based on the work by Stingl et al., that elegantly showed that fluorescently labelled DNA formed periplasmic foci and that their half-life was dependent on their capacity to go be internalised into the cytoplasm by ComEC. We therefore still think that the information on the internalisation of the DNA obtained through the live microscopy experiments reinforces the results presented in Figure 5 and is worth presenting. However, to avoid redundancies we agree to do so as Supplementary material. The text of the results section, as well as figure 6 were changed accordingly.

9. Method section: Information about NikR and MotB-tagging is missing. For ease of reading, please sort the Method section starting with all genetic constructions, separated in H. pylori and E. coli, biochemical assays and bacterial assays. Also information about the Western blots, including the used antibodies is missing.

We thank this reviewer for pointing out the missing information on the Western blots. We have now added it in the new version of the manuscript. We have also changed the order of the Methods.

I highly recommend this manuscript for publication if the points above have been addressed by the authors. The results will substantially contribute to higher understanding of this extremely important topic of genetic exchange in H. pylori.

Once again we thank this reviewer for his/her comments.

Reviewer 2

Damke et al describe that ComH, which shows a periplasmic location, is essential for the import of dsDNA into the periplasm, but it is not involved in the internalization of ssDNA into the cytoplasm. The manuscript resolves many of the puzzles of the DNA uptake machinery of H. pylori competent cells, but before publication the authors must solve the problem with the DNA binding assays.

General comments

- 1. Page (P3), line (L) 6-7. I recommend not to quote the statement that “H. pylori imports up to 1.6 Mb of transforming DNA into the periplasm” until any other independent confirmation is reported. I have some problems with the numbers reported in the original paper. Since the H. pylori type-IV secretion system imports dsDNA into the periplasmic space at a velocity of 1,300-bp/s, and it is expected that ComEC transport ssDNA through the inner membrane at a velocity of ~80-bp/s, we have to assume that dsDNA has to accumulate into the periplasm. H. pylori is 3 µm long with a diameter of about 0.5 µm and an expected total volume of the periplasm of ~4 x 10E-16 L. Perhaps the information is correct, but to accept that up ~5 mm of linear dsDNA (1.6 Mb) of tDNA can accumulate into the busy periplasm an independent confirmation is required.*

We have now changed the sentence in the introduction.

- 2. P3, L 12 from bottom. The sentence is misleading, absence of ComH reduced DNA uptake rather than the recombination frequency. Inactivation of comEC leads to ~2 x 10E-9 transformants or spontaneous mutants in its target sequence. I wonder what is the spontaneous mutation rate in specific H. pylori strains? I thought that mutations are on average more frequent than recombination events on nature.*

We agree with this reviewer that using “recombination frequencies” might suggest a defect in the recombination process itself. We have now changed the text mentioning “yield of recombinant clones” to stick to the experimental observation. We have now determined the spontaneous mutation frequencies for streptomycin resistance and mention them in the text. We have also added the corresponding Methods description.

- 3. P4, L 13-14. In the absence of comH ~1% of cells accumulate fluorescently labelled dsDNA or ~20% of total dsDNA (donor/recipient rate) is accumulated in the*

periplasmic space, but the transformation frequency drops >400.000-fold. Please elaborate on such discrepancies.

While there is a general correlation between the proportion of cells with foci and the transformation capacity, the former's sensitivity and range of linearity are much smaller. Moreover, we cannot rule out a certain degree of non-specific binding and protection from DNases of the DNA on the cell surface that would explain the PCR results. We have now added a sentence in the results section to address this point. The genetic approach – measuring transformation efficiency- is a much more sensitive one. Finally, ComH impairment could affect not only the accumulation of tDNA in the periplasm but, as shown later on in our manuscript, also the passage of the DNA through the inner membrane. The effect on transformation would then be a cumulative one as explained in the Discussion section.

4. P5, L11-12 from bottom). From the data presented in Fig. 2b, there is a discrepancy in the ratio of colocalization of DNA (4 green dots, one faint) with ComH (8 red dots, and >6 faint dots). I can see that 100% of the tDNA foci colocalized with the ones of ComH.

What our experiments show is the presence of ComH-Cherry clusters at the sites of fluorescent DNA foci. As explained in the new version of our manuscript, this was confirmed quantitatively by the image analysis using the JACOP plugin of ImageJ. The excess of ComH-Cherry clusters could be explained by the presence of non-labelled DNA issued from bacterial lysis. Alternatively, ComH could spontaneously form clusters in the absence of DNA. At this point we do not have the means to make sure there is no DNA at the sites of ComH-Cherry clusters. All this is now discussed in the new version of the manuscript.

5. P6, L8. Are competent comH-His variants proficient in natural transformation? The microscale thermophoresis data revealed a very high dispersion rate (Fig. 3c), I wonder how the authors reach a Kd value.

The ComH-His variants were only expressed in *E. coli*. The aim was to purify the proteins and use them for the biochemical characterisation experiments.

We have now repeated the microscale thermophoresis experiments with larger amounts of protein and obtained more precise Kd value for dsDNA. We performed titration with higher final concentration of protein and obtained a well-defined plateau at high concentration. Please refer to the new Figures 3C and the new Kd value in the text.

6. P6, L11-12. The authors assumption “The fusion protein MBP-ComH exhibits a slightly higher affinity compared to ComH-His6 probably due to a stabilizing effect of the MBP protein” jeopardized all DNA binding assays. Alternatively, the presence of His-tag misfolds the ComH protein, and ComH-His is a mutant variant. There are many papers describing this, even rendering fused proteins more thermosensitive. MBP-ComH binds ssDNA with lower affinity compared to dsDNA, but ComH-His does not bind ssDNA.

We agree with this reviewer that it is not always obvious to compare proteins fused with different tags. However, our results using different fusions and various experimental techniques show consistently a preference of ComH for dsDNA when compared to ssDNA. Actually, MST experiments indicated that there is some non-specific binding by MBP that is not detected by the less sensitive EMSA. We therefore have used the His fusions for obtaining Kd values using both SV-AUC and MST approaches. It is worth mentioning here that using MST we do see some residual binding to ssDNA.

7. P7, L14-15. The authors stated that “The Kd obtained for ComH-CTD (2.9 μM) (Fig. 4d) was comparable to that obtained for the full-length protein (1.25 μM) (Fig. 3c). However, from the data show in Fig. 4c we can deduce a Kd of ~0.4 μM for ComH-CTD-His and of ~0.75 μM for full-length ComH-His. Furthermore, it can be deduced a Kd of ~0.1 μM for both MBP-ComH-CTD and full-length MBP-ComH (Fig. S4b). Please remove the tag or focus in the MBP-ComH variants.

The Kd values obtained from the experiments presented in the new Fig. 3c for the full-length protein indicate that the CTD does not bind DNA as efficiently as the full-length protein. We have now addressed this in the new version.

8. P7, L14. With the protein concentration used, the authors cannot reach a conclusion about the substrate preference.

As mentioned in point 5 the MST experiment was repeated with a broader range of protein concentrations.

Specific comments

P2, L16. Please correct. The authors stated that ComE(A) is conserved amongst most naturally transformable Gram-positive and Gram-negative bacteria, but, although conserved, in some bacteria ComEA is membrane bound, whereas in others ComE or ComE(A) is free in the cytoplasm.

We agree on the fact that some ComE(A) proteins, in particular amongst Gram-positive bacteria, are associated with the cell membrane, while in the Gram-negative bacteria where this was analysed, the protein is found in the periplasm. To the best of our knowledge, there are no reports of *bona fide* ComE(A) proteins involved in competence and present in the cytoplasm.

Reviewer 3

*When it comes to natural competence for transformation, the human pathogen *H. pylori* is particular and different from most other studied competent Gram-negatives, as it uses a type IV secretion system instead of a type IV pilus as central part of its DNA-uptake machinery. As such, an open question in the field has been as to which protein would accept the incoming genetic material in the periplasm of this organism, given that a homolog of the DNA-binding protein ComE/ComEA is missing. In this study, Damke et al characterized a recently identified competence protein (Ref#26), ComH, and provide evidence that this protein is responsible for DNA import into the periplasm. In addition the protein interacts with the inner membrane DNA translocator ComEC though this interaction has not been addressed on the mechanistic level.*

The study is well performed and combines bacterial genetics, basic biochemistry, and imaging. Indeed, the authors used well thought-through experimental designs such as the ssDNA electroporation assay or a PCR-based DNA uptake assay to provide convincing data that helped identifying the step in which ComH acts in the DNA uptake process. They further investigated this finding - mostly through in vitro assays.

The first part of the paper (KO, complementation, mCherry fusion, DNA co-location, purification and EMSA) is well performed and very clear but somewhat recapitulates previous studies on ComEA (e.g., ref#6). The second part of the paper is novel in a way that it identified separate regions of the ~55kDa ComH protein (much larger than the known small ComEA proteins of 10-15kDa of other Gram-negative bacteria) and assigns these regions to specific functions. Precisely, the C-terminal domain was shown to bind to DNA while the N-terminal domain was shown to bind to the inner membrane DNA translocation channel ComEC.

Overall, this is a conclusive study that will be of importance for the natural transformation community and probably also beyond.

Below I provide suggestions on how to further improve the manuscript.

We also thank this reviewer for her/his positive and constructive comments on our work.

- While the interaction of ComH with dsDNA is shown with 3-4 different methods, the interaction with ComEC would benefit from an additional approach to support the data (e.g., pull-down from native cell lysates, BTH or similar approaches). Indeed, with the current approach both proteins are synthesized in *E. coli* and then mixed. But how would this look like in the periplasm?*

Following the suggestions of this reviewer we have attempted several approaches to confirm the interaction in *H. pylori* cells. For that we have tagged ComEC with either FLAG, GFP or mCherry on its own locus but in no case we were able to detect the protein by Western blot or microscopy. Similar problems in other bacterial species, probably due to the low expression of ComEC proteins, have been reported by other labs. We have also tried to express in the periplasm the OB domain fused to an HA tag by adding a peptide signal sequence but again we couldn't detect the protein.

- *At this stage, it is also theoretically possible that the purified ComH already binds DNA from the E. coli cytoplasm (as was the case for ComEA; ref#6) and that the OB-binding domain of ComEC binds to the same DNA and not directly to ComH. Also the Histag might bind to the DNA (as for example speculated by Balaban et al 2014, who showed unspecific DNA binding through His-tagged proteins as a result of electrostatic charges/interactions). This should be addressed (e.g., by adding nucleases to the pull-down assay).*

To address this point we have followed the reviewer's suggestion and repeated the pulldown experiments and adding a nuclease. This is now presented in Supplementary Fig. 9 and referred to in the main text.

- *Fig. 2B: The authors wrote "Two ComH-mCherry distribution patterns were observed. While the majority of the cells showed a rather homogenous distribution of the protein, in about a third of the cells it formed distinct clusters (Fig. 2b)." This is not really reflected in the images. The protein doesn't seem to localize homogenously but in distinctive regularly interspaced puncta. There is also no obvious periplasmic localization. This inconsistency should be addressed by the authors.*

To address this issue we have now changed the text in our new version of the manuscript and detailed the way we analysed the co-localisation between ComH clusters and tDNA foci.

- *The discussion seems more a summary of the results than a global picture. I especially miss the upstream part => how do the authors see the interaction of the ComH protein with the T4SS? In this context, the manuscript would be significantly strengthened if the authors would experimentally address this question and visualize the localization of ComH in the presence and absence of the T4SS and check if there is any co-localisation with membrane-proteins of the T4SS.*

The point raised here is indeed a very interesting one. Our preliminary experiments showed that the absence of ComB2 did not affect the distribution pattern of ComH-Cherry. However, a more detailed study, including the inactivation of other components of the ComB uptake machinery are required. While we plan to address this kind of analysis we still don't have the tools to visualise by microscopy the ComB proteins, thus limiting the conclusions to be drawn from those experiments. We have now added a sentence in the Discussion section to comment on this point.

Minor points:

- *Please describe genetic constructs (e.g., page 5: $\Delta comH rdxA::comH-FLAG$ => why in *rdxA* gene? Neutral locus?)*

rdxA is indeed a non-essential gene that does not affect growth. The description is now included in the text.

- *Page 4: can the authors speculate on the residual 1% of periplasmic tDNA foci in the periplasm?*

As mentioned in the response to Reviewer 2, while there is a general correlation between the proportion of cells with foci and the transformation capacity, the former's sensitivity and range of linearity are much smaller. Moreover, we cannot rule out a certain degree of non-specific binding and protection from DNases of the DNA on the cell surface that would explain the PCR results. We have now added a sentence in the results section to address this point. The genetic approach –measuring transformation efficiency- is a much more sensitive one. Finally, ComH impairment could affect not only the accumulation of tDNA in the periplasm but, as shown later on in our manuscript, also the passage of the DNA through the inner membrane. The effect on transformation would then be a cumulative one as explained in the Discussion section.

- *The authors should ensure that references are properly used (e.g., citing the references that fit with the organism described in the preceding sentence).*

We have now revised the reference to make sure they corresponded to the text in the sentence.

- *Page 9 “In *N. gonorrhoeae* and *V. cholerae* it is proposed that the incoming DNA is pulled across the outer membrane by the periplasmic DNA binding protein ComEA (ComE in *Neisseria*) => needs references (Ref#5 to 7).*

We have now modified the corresponding sentence and added the references.

- *Page 10: “The lack of sequence specificity in the binding of ComH to dsDNA is consistent with the previous observation that *H. pylori* displays no sequence bias for DNA uptake.” => this sentence is misleading as it implies that ComEA/ComE would exert such a sequence-specificity in those Gram-negatives that distinguish DNA and required DNA uptake sequences (DUS/USS) for import. However, this is not the case, as DNA sorting occurs outside the cell (for example through ComP in *Neisseria meningitidis*; Pelicic group).*

We completely agree with this reviewer. The sentence does suggest that the ComEA proteins contribute to the selection of DUS/USS harbouring DNA for import. At no point we meant to imply that. It was an unfortunate choice. We have now removed the sentence.

- *As the authors provide a quantification of panel c in panel d of Fig.1, the same should be done for Fig. 5d.*

The graph showing the quantification is now added.

- *Legend figure 7 needs reference.*

References have been incorporated.

REVIEWERS' COMMENTS:

Reviewer #1 (Remarks to the Author):

The authors answered most of the raised questions. One single comment to be addressed:

The authors' answer to my previous point 5 is highly interesting in terms of understanding ComH function in natural transformation. The authors mention that within 7 min, DNA is sufficiently taken up in order to evaluate competence development BUT that ComH m-Cherry clusters did form later. At timepoint 7 min, only 5 % of the cells exhibited distinguishable ComH m-Cherry clusters. After 90 min, the authors maximally detected ComH-Cherry clusters – a timepoint much later than the completion of the fast uptake over the outer membrane (it is saturated within 10 min, as also the authors observe, since they chose 7 min to monitor the fraction of bacteria with DNA foci). Therefore, interaction between incoming DNA and ComH was maximal AFTER pulling DNA over the outer membrane (this observation should be mentioned somewhere).

I still doubt from the kinetics of outer membrane import of DNA that protein binding to DNA generates the pulling force for this macromolecule to traverse the outer membrane in *H. pylori*. In bacteria with ComE, kinetics are different. An alternative role of ComH as periplasmic DNA receptor (which the authors undoubtedly showed with very solid experiments in an elegant way) could be to prevent extensive uptake of the huge amount of periplasmic DNA into the cytoplasm (forming clusters later to trap excessive periplasmic DNA). In this scenario, ComH might activate ComB complexes in a so far unknown way but is not directly involved in DNA uptake. Of course, this hypothesis awaits more evidence in future studies.

Anyhow, the overall manuscript is highly interesting and further analysis of ComH function is beyond the scope here. However, it would help the reader to get more sophisticated information about ComH function if the authors insert the results mentioned in their answer to my point 5 into the final manuscript. I recommend to show an additional microscopic photo of ComH m-Cherry after 7 min in Fig. 2b), mentioning that DNA was taken up within this time window, implicating a more complex role of ComH than suggested for ComE homologues.

I'm looking forward to the publication of this manuscript. It was a pleasure to read it!

Reviewer #2 (Remarks to the Author):

The authors have answer clearly to all my queries. I consider that the results are novel and important for the understanding of the mechanism of uptake of environmental DNA in natural competent *H. pylori*.

Reviewer #3 (Remarks to the Author):

The authors have done a very thorough job in revising their manuscript and in properly addressing the reviewers' concerns. They also included information concerning limitation of the study/the methodology where needed and presented their findings in a more cautious manner. For this reason, the manuscript has improved from an already very good initial version.

I'd like to congratulate the authors for this excellent study, which I strongly believe will be of great importance for our general understanding of natural competence for transformation in bacteria.

M. Blokesch

Response to reviewers

We thank the reviewers for their positive evaluation of our latest version of the manuscript. Their comments on our original submission provided an invaluable help for its improvement. The peer review process has been extremely useful.

REVIEWERS' COMMENTS:

Reviewer #1 (Remarks to the Author):

The authors answered most of the raised questions. One single comment to be addressed:

The authors' answer to my previous point 5 is highly interesting in terms of understanding ComH function in natural transformation. The authors mention that within 7 min, DNA is sufficiently taken up in order to evaluate competence development BUT that ComH m-Cherry clusters did form later. At timepoint 7 min, only 5 % of the cells exhibited distinguishable ComH m-Cherry clusters. After 90 min, the authors maximally detected ComH-Cherry clusters – a timepoint much later than the completion of the fast uptake over the outer membrane (it is saturated within 10 min, as also the authors observe, since they chose 7 min to monitor the fraction of bacteria with DNA foci). Therefore, interaction between incoming DNA and ComH was maximal AFTER pulling DNA over the outer membrane (this observation should be mentioned somewhere).

*I still doubt from the kinetics of outer membrane import of DNA that protein binding to DNA generates the pulling force for this macromolecule to traverse the outer membrane in *H. pylori*. In bacteria with ComE, kinetics are different. An alternative role of ComH as periplasmic DNA receptor (which the authors undoubtedly showed with very solid experiments in an elegant way) could be to prevent extensive uptake of the huge amount of periplasmic DNA into the cytoplasm (forming clusters later to trap excessive periplasmic DNA). In this scenario, ComH might activate ComB complexes in a so far unknown way but is not directly involved in DNA uptake. Of course, this hypothesis awaits more evidence in future studies.*

Anyhow, the overall manuscript is highly interesting and further analysis of ComH function is beyond the scope here. However, it would help the reader to get more sophisticated information about ComH function if the authors insert the results mentioned in their answer to my point 5 into the final manuscript. I recommend to show an additional microscopic photo of ComH m-Cherry after 7 min in Fig. 2b), mentioning that DNA was taken up within this time window, implicating a more complex role of ComH than suggested for ComE homologues.

We agree with this reviewer on the need to further explore the role of ComH in view of the surprising observation that the protein clusters increase for up to 90 minutes. To foster further studies, we have now added a sentence in the Results section highlighting this observation (page 6 of the manuscript).

I'm looking forward to the publication of this manuscript. It was a pleasure to read it!

Reviewer #2 (Remarks to the Author):

The authors have answer clearly to all my queries. I consider that the results are novel and important for the understanding of the mechanism of uptake of environmental DNA in natural competent H. pylori.

Reviewer #3 (Remarks to the Author):

The authors have done a very thorough job in revising their manuscript and in properly addressing the reviewers' concerns. They also included information concerning limitation of the study/the methodology where needed and presented their findings in a more cautious manner. For this reason, the manuscript has improved from an already very good initial version.

I'd like to congratulate the authors for this excellent study, which I strongly believe will be of great importance for our general understanding of natural competence for transformation in bacteria.

M. Blokesch

Once again we thank the reviewers for all their feedback and their positive and encouraging comments. It has been a pleasure to interact with them.